# Modelling the response to vaccine in non-human primates to define SARS-CoV-2 mechanistic correlates of protection

Marie Alexandre[1], Romain Marlin[2†], Mélanie Prague[1†], Severin Coleon[3,4], Nidhal Kahlaoui[2], Sylvain Cardinaud[3,4], Thibaut Naninck[2], Benoit Delache[2], Mathieu Surenaud[3,4], Mathilde Galhaut[2], Nathalie Dereuddre-Bosquet[2], Mariangela Cavarelli[2], Pauline Maisonnasse[2], Mireille Centlivre[3,4], Christine Lacabaratz[3,4], Aurelie Wiedemann[3,4], Sandra Zurawski[5], Gerard Zurawski[5], Olivier Schwartz[3,6,7], Rogier W Sanders[8], Roger Le Grand[2], Yves Levy[3,4,9], Rodolphe Thiébaut[1,3,10]*

[1]University of Bordeaux, Department of Public Health, Inserm Bordeaux Population Health Research Centre, Inria SISTM, Bordeaux, France; [2]Center for Immunology of Viral, Auto-immune, Hematological and Bacterial Diseases (IMVA-HB/IDMIT), Université Paris-Saclay, Inserm, CEA, Fontenay-aux-Roses, France; [3]Vaccine Research Institute, Créteil, France; [4]Inserm U955, Créteil, France; [5]Baylor Scott and White Research Institute, Dallas, United States; [6]Virus & Immunity Unit, Department of Virology, Institut Pasteur, Paris, France; [7]CNRS UMR 3569, Paris, France; [8]Department of Medical Microbiology, Amsterdam UMC, University of Amsterdam Amsterdam Infection & Immunity Institute, Amsterdam, Netherlands; [9]AP-HP, Hôpital Henri-Mondor Albert-Chenevier, Service d'Immunologie Clinique et Maladies Infectieuses, Créteil, France; [10]CHU Bordeaux, Department of Medical information, Bordeaux, France

*For correspondence: rodolphe.thiebaut@u-bordeaux.fr

†These authors contributed equally to this work

Competing interest: The authors declare that no competing interests exist.

**Abstract** The definition of correlates of protection is critical for the development of next-generation SARS-CoV-2 vaccine platforms. Here, we propose a model-based approach for identifying mechanistic correlates of protection based on mathematical modelling of viral dynamics and data mining of immunological markers. The application to three different studies in non-human primates evaluating SARS-CoV-2 vaccines based on CD40-targeting, two-component spike nanoparticle and mRNA 1273 identifies and quantifies two main mechanisms that are a decrease of rate of cell infection and an increase in clearance of infected cells. Inhibition of RBD binding to ACE2 appears to be a robust mechanistic correlate of protection across the three vaccine platforms although not capturing the whole biological vaccine effect. The model shows that RBD/ACE2 binding inhibition represents a strong mechanism of protection which required significant reduction in blocking potency to effectively compromise the control of viral replication.

## Editor's evaluation

This work should be of interest to a broad readership in infectious diseases, especially those people interested in modeling of infections. It combines statistical and mechanistic modeling to find assayable correlates of immunity for vaccines. This method could be relevant to many diseases or vaccines, although the particular markers identified here likely will be limited in their generalizability.

## Introduction

There is an unprecedented effort for SARS-CoV-2 vaccine development with 294 candidates currently evaluated (*World Health Organization, 2021*). However, variants of concern have emerged before the vaccine coverage was large enough to control the pandemics (*Cobey et al., 2021*). Despite a high rate of vaccine protection, these variants might compromise the efficacy of current vaccines (*Kuzmina et al., 2021*; *Planas et al., 2021*; *Lustig et al., 2021*; *Zhou et al., 2021*). Control of the epidemic by mass vaccination may also be compromised by unknown factors such as long-term protection and the need of booster injections in fragile, immuno-compromised, elderly populations, or even for any individual if protective antibody levels wane. Furthermore, the repeated use of some of the currently approved vaccine could be compromised by potential adverse events or by immunity against vaccine viral vectors (*Greinacher et al., 2021*). Finally, the necessity to produce the billions of doses required to vaccinate the world's population also explains the need to develop additional vaccine candidates.

The identification of correlates of protection (CoPs) is essential to accelerate the development of new vaccines and vaccination strategies (*Koch et al., 2021*; *Jin et al., 2021*). Binding antibodies to SARS-CoV-2 and in vitro neutralization of virus infection are clearly associated with protection (*Khoury et al., 2021*; *Yu et al., 2020*; *Earle et al., 2021*; *Feng et al., 2021*). However, the respective contribution to virus control in vivo remains unclear (*Zost et al., 2020*), and many other immunological mechanisms may also be involved, including other antibody-mediated functions (antibody-dependent cellular cytotoxicity [ADCC], antibody-dependent complement deposition [ADCD], antibody-dependent cellular phagocytosis [ADCP]; *Yu et al., 2020*; *Mercado et al., 2020*; *Tauzin et al., 2021*), as well as T cell immunity (*McMahan et al., 2021*). Furthermore, CoP may vary between the vaccine platforms (*Plotkin, 2013*; *Plotkin, 2020*; *Bradfute and Bavari, 2011*; *Dagotto et al., 2020*).

Non-human primate (NHP) studies offer a unique opportunity to evaluate early markers of protective response (*Muñoz-Fontela, 2020*; *Eyal and Lipsitch, 2021*). Challenge studies in NHP allow the evaluation of vaccine impact on the viral dynamics in different tissue compartments (upper and lower respiratory tract) from day 1 of virus exposure (*Yu et al., 2020*; *Mercado et al., 2020*; *Corbett et al., 2020*). Such approaches in animal models may thus help to infer, for example, the relation between early viral events and disease or the capacity to control secondary transmissions.

Here, we propose to apply a model-based approach on NHP studies to evaluate (i) the immune mechanisms involved in the vaccine response and (ii) the markers capturing this/these effect(s) leading to identification of mechanisms of protection and definition of mechanistic CoP (*Plotkin and Gilbert, 2012*). First, we present a mechanistic approach based on ordinary differential equation (ODE) models reflecting the virus-host interaction inspired from models proposed for SARS-CoV-2 infection (*Gonçalves et al., 2020*; *Kim et al., 2021*; *Gonçalves et al., 2021*; *Wang et al., 2020*; *Marc et al., 2021*; *Ke et al., 2021*) and other viruses (*Myers et al., 2021*; *Baccam et al., 2006*; *Goyal et al., 2019*; *Goyal et al., 2017*). The proposed model includes several new aspects refining the modelling of viral dynamics in vivo, in addition to the integration of vaccine effect. A specific inoculum compartment allows distinguishing the virus coming from the challenge inoculum and the virus produced de novo, which is a key point in the context of efficacy provided by antigen-specific pre-existing immune effectors induced by the vaccine. Then, an original data mining approach is implemented to identify the immunological biomarkers associated with specific mechanisms of vaccine-induced protection.

We apply our approach to a recently published study (*Marlin et al., 2021*) testing a protein-based vaccine targeting the receptor-binding domain (RBD) of the SARS-CoV-2 spike protein to CD40 (αCD40.RBD vaccine). Targeting vaccine antigens to dendritic cells via the surface receptor CD40 represents an appealing strategy to improve subunit-vaccine efficacy (*Flamar et al., 2012*; *Zurawski et al., 2017*; *Cheng et al., 2018*; *Godot et al., 2020*) and for boosting natural immunity in SARS-CoV-2 convalescent NHP.

We show that immunity induced by natural SARS-CoV-2 infection, as well as vaccine-elicited immune responses contribute to viral load control by (i) blocking new infection of target cells and (ii) by increasing the loss of infected cells. The modelling showed that antibodies inhibiting binding of RBD to ACE2 correlated with blockade of new infections and RBD-binding antibodies correlate with the loss of infected cells, reflecting importance of additional antibody functionalities. The role of RBD/ACE2-binding inhibition has been confirmed in two other vaccine platforms.

## Results

### A new mechanistic model fits the in vivo viral load dynamics in nasopharyngeal and tracheal compartments

The mechanistic model aims at capturing the viral dynamics following challenge with SARS-CoV-2 virus in NHP. For that purpose, we used data obtained from 18 cynomolgus macaques involved in the vaccine study reported by *Marlin et al., 2021*, and exposed to a high dose ($1 \times 10^6$ pfu) of SARS-CoV-2 administered via the combined intra-nasal and intra-tracheal route. The viral dynamics during the primary infections were characterized by a peak of genomic RNA (gRNA) production 3 days post-infection in both tracheal and nasopharyngeal compartments, followed by a decrease toward undetectable levels beyond day 15 (*Figure 1—figure supplement 1*). At the convalescent phase (median 24 weeks after the primary infection), 12 macaques were challenged with SARS-CoV-2 a second time, 4 weeks after being randomly selected to receive either a placebo (*n*=6) or a single injection of the αCD40.RBD vaccine (*n*=6) (*Figure 1A*). A third group of six naïve animals were infected at the same time. Compared to this naïve group, viral dynamics were blunted following the second challenge of convalescent animals with the lowest viral load observed in vaccinated animals (*Figure 1B*, *Figure 1—figure supplement 2*).

We developed a mathematical model to better characterize the impact of the immune response on the viral gRNA and subgenomic RNA (sgRNA) dynamics, adapted from previously published work (*Gonçalves et al., 2020*; *Kim et al., 2021*; *Baccam et al., 2006*), which includes uninfected target cells (*T*) that can be infected ($I_1$) and produce virus after an eclipse phase ($I_2$). The virus generated can be infectious ($V_i$) or non-infectious ($V_{ni}$). Although a single compartment for de novo produced viruses (*V*) could be mathematically considered, two distinct ODE compartments were assumed for a better understanding of the model. We completed the model by a compartment for the inoculum to distinguish between the injected virus ($V_s$) and the virus produced de novo by the host ($V_i$ and $V_{ni}$). In both compartments of the upper respiratory tract (URT), the trachea and nasopharynx, viral dynamics were distinctively described by this model (*Figure 2A*). Viral exchange between the two compartments was tested (either from the nasopharynx to the trachea or vice versa). However, as described in the literature (*Gonçalves et al., 2021*; *Ke et al., 2020*; *Pinky et al., 2021*) and demonstrated by the additional modelling work in Appendix 1 'Model building', viral transport within the respiratory tract plays a negligible role in viral kinetics compared with viral clearance. Consequently, no exchange was considered in the model. Using the gRNA and sgRNA viral loads, we jointly estimated (i.e., shared random effects and covariates) the viral infectivity (*β*), the viral production rate (*P*), and the loss rate of infected cells (*δ*) in the two compartments. We assumed that gRNA and sgRNA were proportional to the free virus and the infected cells, respectively. This modelling choice relied on both biological and mathematical reasons (see section Materials and methods for more details). Due to identifiability issues, the duration of the eclipse phase (1/*k*), the clearance of free viruses from the inoculum ($c_i$) and produced de novo (*c*) were estimated separately by profile likelihood and assumed to be identical in the two compartments of the URT. In addition, infectious and non-infectious viruses were assumed to be cleared at the same rate. We estimated the viral infectivity at $0.95 \times 10^{-6}$ (CI$_{95\%}$ [$0.18 \times 10^{-6}$; $4.94 \times 10^{-6}$]) (copies/mL)$^{-1}$ day$^{-1}$ in naïve animals, which is in the range of previously reported modelling results whether in the case of SARS-CoV-2 virus (*Kim et al., 2021*; *Wang et al., 2020*) or influenza (*Myers et al., 2021*; *Baccam et al., 2006*). We found estimates of the loss rates of infected cells of 1.04 (CI$_{95\%}$ [0.79; 1.37]) day$^{-1}$, corresponding to a mean half-life of 0.67 day. This estimation was consistent with previously published results obtained on SARS-CoV-2 virus showing the mean value of this parameter ranging from 0.60 to 2 day$^{-1}$ (i.e., half-life between 0.35 and 1.16 days) (*Gonçalves et al., 2020*; *Kim et al., 2021*; *Gonçalves et al., 2021*; *Wang et al., 2020*; *Marc et al., 2021*). The eclipse phase (3 day$^{-1}$) was found similar to the values commonly used in the literature (*Gonçalves et al., 2020*; *Marc et al., 2021*; *Myers et al., 2021*; *Baccam et al., 2006*). Here, we distinguished the clearance of the inoculum which was much higher (20 virions day$^{-1}$) as compared to the clearance of the virus produced de novo (3 virions day$^{-1}$). While the half-life of the virus de novo produced usually approximates 1.7 hr (i.e., *c*=10 day$^{-1}$) (*Gonçalves et al., 2020*; *Gonçalves et al., 2021*; *Marc et al., 2021*; *Myers et al., 2021*), because of this distinction, our model provided a higher estimation of 5.5 hr which remained in accordance with the estimations obtained by *Baccam et al., 2006*, on influenza A. Furthermore, the viral production by each infected cells was estimated to be higher in the nasopharyngeal compartment ($12.1 \times 10^3$ virions cell$^{-1}$ day$^{-1}$, CI$_{95\%}$ [$3.15 \times 10^3$; $46.5 \times 10^3$]) as

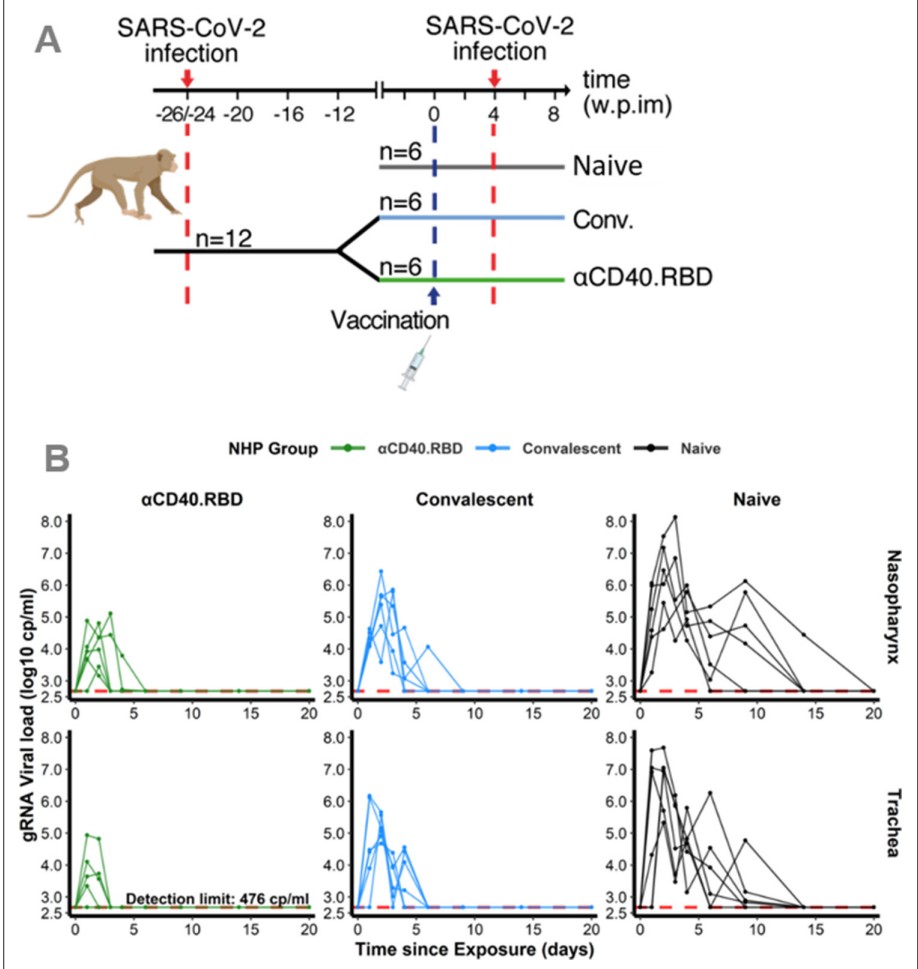

**Figure 1.** Design of the study 1 and viral dynamics. (**A**) *Study design*. Cynomolgus macaques (*Macaca fascicularis*), aged 37–58 months (8 females and 13 males). 24–26 weeks post-infection with SARS-CoV-2, 12 of these animals were randomly assigned in two experimental groups. The convalescent-vaccinated group (*n*=6) received 200 µg of αCD40.RBD vaccine. The other six convalescent animals were used as controls. Additional six age matched (43.7 months±6.76) cynomolgus macaques from same origin were included in the study as controls naïve from any exposure to SARS-CoV-2. Four weeks after immunization, all animals were exposed to a total dose of $10^6$ pfu of SARS-CoV-2 virus via the combination of intra-nasal and intra-tracheal routes. In this work, only data collected from the second exposure were considered. (**B**) Individual $\log_{10}$ transformed genomic RNA (gRNA) viral load dynamics in nasopharyngeal swabs (top) and tracheal swabs (bottom) after the initial exposure to SARS-CoV-2 in naïve macaques (black, right) and after the second exposure in convalescent (blue, middle) and αCD40.RBD-vaccinated convalescent (green, left) groups. Horizontal red dashed lines indicate the limit of quantification.

The online version of this article includes the following source data and figure supplement(s) for figure 1:

**Source data 1.** Genomic RNA (gRNA) viral load longitudinally measured in the trachea and nasopharynx after the second exposure in the study 1.

**Source data 2.** Genomic RNA (gRNA) viral load longitudinally measured in the trachea and nasopharynx after the first exposure for convalescent non-human primates (NHPs) in the study 1.

**Source data 3.** Anti-spike IgG longitudinally measured post-immunization and quantified by Luminex in the study 1.

**Source data 4.** Quantification of the spike/ACE2-binding inhibition longitudinally measured post-immunization and quantified by Mesoscale Discovery (MSD) assay (in 1/ECL) in the study 1.

**Source data 5.** Anti-N and anti-RBD binding antibodies longitudinally measured post-immunization and quantified by Mesoscale Discovery (MSD) assay (in AU mL$^{-1}$) in the study 1.

**Source data 6.** Subgenomic RNA (sgRNA) viral load longitudinally measured in the trachea and nasopharynx after

*Figure 1 continued on next page*

*Figure 1 continued*

the second exposure in the study 1.

**Source data 7.** Antigen-specific T-cell response longitudinally measured post-exposure in % of CD4+ T cells measured by ICS in the study 1.

**Source data 8.** Antigen-specific T-cell response longitudinally measured post-exposure in % of CD8+ T cells measured by ICS in the study 1.

**Source data 9.** T-cell response expressing IFN-γ longitudinally measured post-exposure by ELISpot in the study 1.

**Source data 10.** Cytokine concentrations measured post-exposure in the study 1.

**Source data 11.** Quantification of the neutralization function of antibodies against three variants (B117, B1351, and D614G) longitudinally measured post-exposition (in ED50) in the study 1.

**Figure supplement 1.** Viral dynamics after the first exposure to SARS-CoV-2 and biomarker measurements from the first to the second exposure to SARS-CoV-2.

**Figure supplement 2.** Subgenomic viral dynamics after the second exposure to SARS-CoV-2.

**Figure supplement 3.** Antibody measurements after the second exposure to SARS-CoV-2.

**Figure supplement 4.** Antigen-specific T-cell responses in non-human primates (NHPs) after the second exposure to SARS-CoV-2.

**Figure supplement 5.** Cytokines and chemokines in the plasma in non-human primates (NHPs) after the second exposure to SARS-CoV-2.

compared to the tracheal compartment ($0.92 \times 10^3$ virions cell$^{-1}$ day$^{-1}$, CI$_{95\%}$ [$0.39 \times 10^3$; $2.13 \times 10^3$]). These estimations are in agreement with the observation of the intense production of viral particles by primary human bronchial epithelial cells in culture (*Robinot et al., 2021*). In particular, they are in the range of estimates obtained within the URT, either in NHP (*Gonçalves et al., 2021*) or in humans (*Wang et al., 2020*), with the product $p \times T_0$ equals to $15.1 \times 10^8$ (CI$_{95\%}$ [$3.98 \times 10^8$; $58.1 \times 10^8$]) and $0.21 \times 10^8$ (CI$_{95\%}$ [$0.088 \times 10^8$; $0.48 \times 10^8$]) virions mL$^{-1}$ day$^{-1}$ in the nasopharynx and the trachea, respectively. By allowing parameters to differ between animals (through random effects), the variation of cell infectivity and of the loss rate of infected cells captured the observed variation of the dynamics of viral load. The variation of those parameters could be partly explained by the group to which the animals belong reducing the unexplained variability of the cell infectivity by 66% and of the loss rate of infected cells by 54% (*Supplementary file 1*). The model fitted well the observed dynamics of gRNA and sgRNA (*Figure 2B*).

## Modelling of the dynamics of viral replication argues for the capacity of αCD40.RBD vaccine to block virus entry into host cells and to promote the destruction of infected cells

We distinguish the respective contribution of the vaccine effect and post-infection immunity on the reduction of the cell infection rate and the increase of the clearance of infected cells. Because blocking de novo infection and promoting the destruction of infected cells would lead to different viral dynamics profile (*Figure 2—figure supplement 1*), we were able to identify the contribution of each mechanism by estimating the influence of the vaccine compared to placebo or naïve animals on each model parameter. The αCD40.RBD vaccine reduced by 99.6% the infection of target cells in the trachea compared to the naïve group. The estimated clearance of infected cells was 1.04 day$^{-1}$ (95% CI 0.75; 1.45) in naïve macaques. It was increased by 80% (1.86 day$^{-1}$) in the convalescent macaques vaccinated by αCD40.RBD or not.

The mechanistic model allows predicting the dynamics of unobserved compartments. Hence, a very early decrease of the target cells (all cells expressing ACE2) as well as of the viral inoculum which fully disappeared from day 2 onward were predicted (*Figure 2C*). In the three groups, the number of infected cells as well as infectious viral particles increased up to day 2 and then decreased. We show that this viral dynamic was blunted in the vaccinated animals leading to a predicted maximum number of infectious viral particles in the nasopharynx and the trachea below the detection threshold (*Figure 2C*). The number of target cells would be decreased by the infection in the naïve and the convalescent groups, whereas it would be preserved in vaccinated animals.

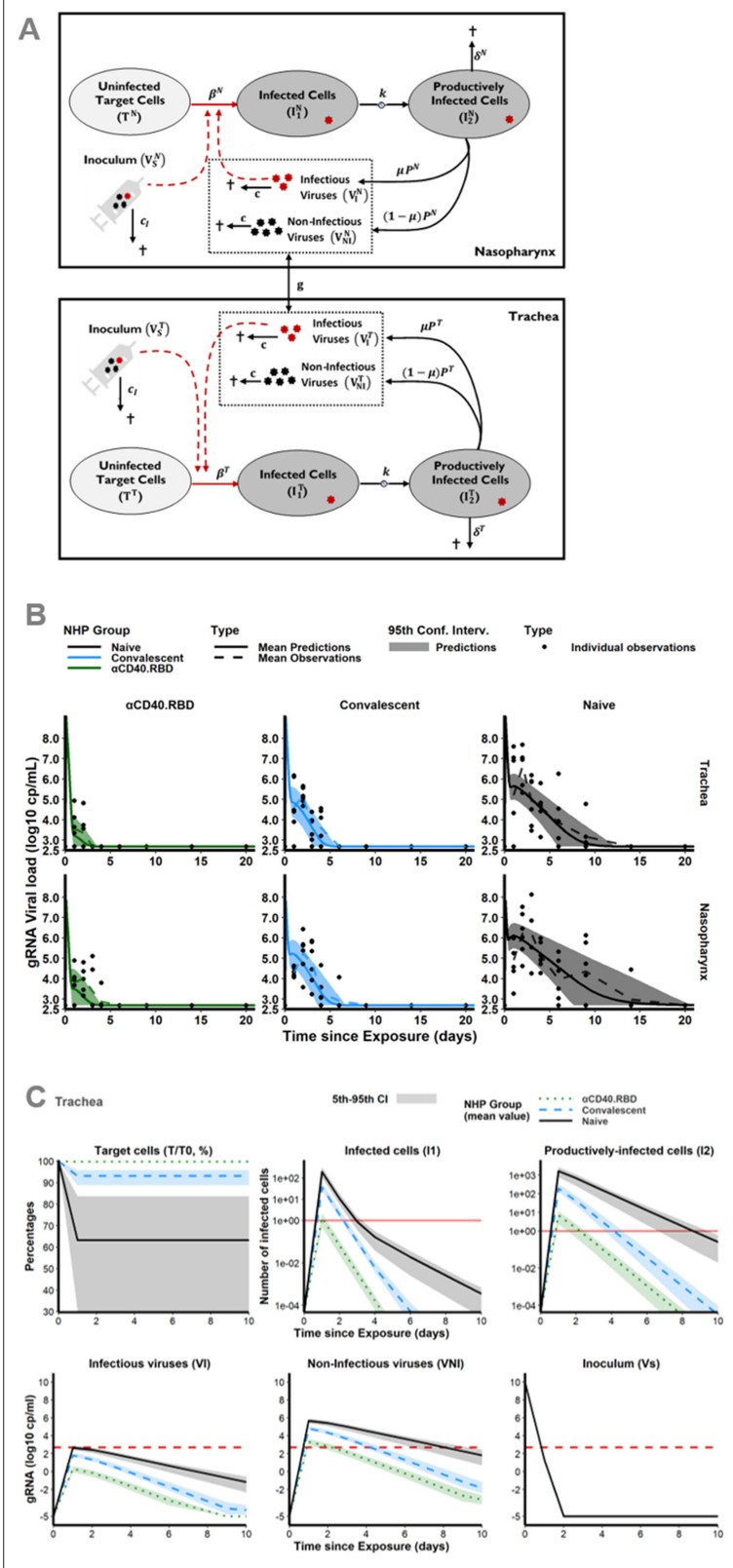

**Figure 2.** Mechanistic modelling. (**A**) Description of the model in the two compartments: the nasopharynx and the trachea. (**B**) Model fit to the $\log_{10}$ transformed observed genomic RNA (gRNA) viral loads in tracheal (top) and nasopharyngeal (bottom) compartments after the initial exposure to SARS-CoV-2 in naïve macaques (black, right) and after the second exposure in convalescent (blue, middle) and vaccinated (green, left) animals. Thick solid and

*Figure 2 continued*

dashed lines indicate mean viral load dynamics predicted and observed, respectively. Shaded areas indicate the 95% confidence intervals of the predictions. Dots represents observations. (**C**) Model predictions of unobserved quantities in the tracheal compartment for naïve (black, solid lines), convalescent (blue, dashed lines) and vaccinated (green, dotted lines) animals: target cells as percentage of the value at the challenge (top, left), infected cells (top, middle), productively infected cells (top, right), inoculum (bottom, right), infectious (bottom, left) and non-infectious virus (bottom, middle). Thick lines indicate mean values over time within each group. Shaded areas indicate the 95% confidence interval. Horizontal dashed red lines indicate the limit of quantification and horizontal solid red lines highlight the threshold of one infected cell.

The online version of this article includes the following source data and figure supplement(s) for figure 2:

**Source data 1.** Volumes of the trachea and nasopharynx, and weights measured at the time of exposure in four non-human primates (NHPs) in the study 1.

**Source data 2.** Weights of the 18 non-human primates (NHPs) in the study 1.

**Source data 3.** Genomic RNA (gRNA) viral load measured in the trachea and nasopharynx in the two additional non-human primates (NHPs) receiving inoculum via intra-gastric and intra-nasal routes.

**Figure supplement 1.** Modelling of the viral dynamics using mechanistic model.

**Figure supplement 2.** Modelling of the dynamics of viral replication.

## The RBD-ACE2-binding inhibition is the main mechanistic CoP explaining the effect of the αCD40.RBD vaccine on new cell infection

In our study (*Marlin et al., 2021*), an extensive evaluation of the immunological response has been performed with quantification of spike-binding antibodies, antibodies inhibiting the attachment of RBD to ACE2, antibodies neutralizing infection, SARS-CoV-2-specific CD4[+] and CD8[+] T cells producing cytokines and serum cytokine levels (*Figure 3*, *Figure 1—figure supplements 3–5*). Therefore, based on our mechanistic model, we investigated if any of these markers could serve as a mechanistic CoP. Such a CoP should be able to capture the effect of the natural immunity following

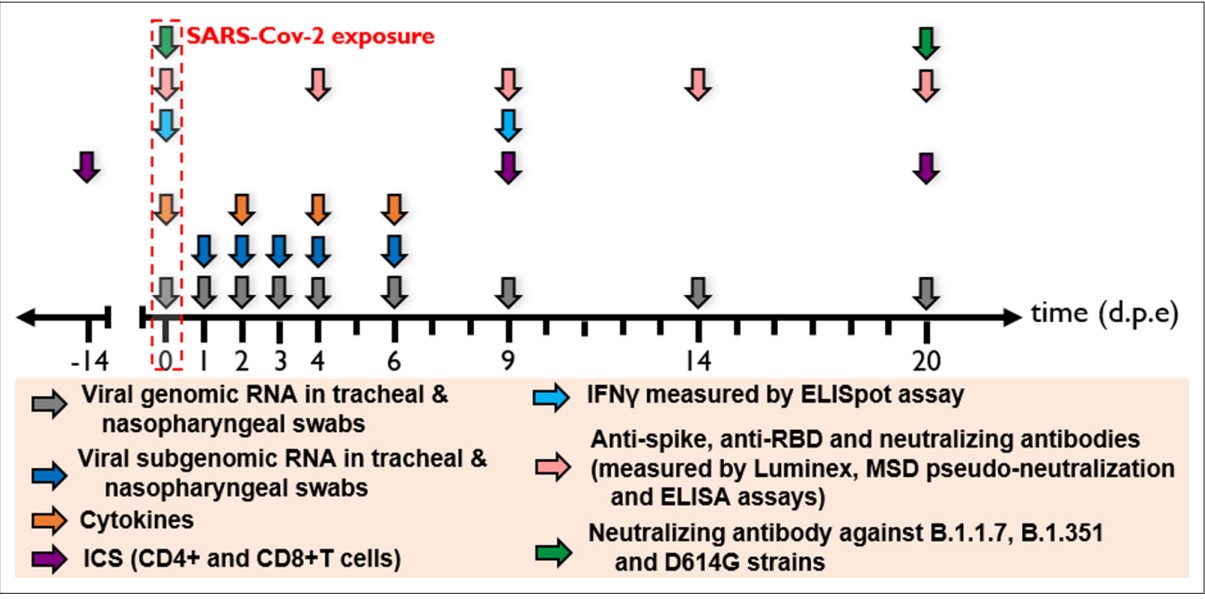

**Figure 3.** Harvest times and measurements. Nasopharyngeal and tracheal fluids were collected at 0, 1, 2, 3, 4, 6, 9, 14, and 20 days post-exposure (d.p.e) while blood was taken at 0, 2, 4, 6, 9, 14, and 20 d.p.e. Genomic and subgenomic viral loads were measured by RT-qPCR. Anti-spike IgG sera were titrated by multiplex bead assay, anti-RBD, and anti-nucleocapside (**N**) IgG were titrated using a commercially available multiplexed immunoassay developed by Mesoscale Discovery (MSD, Rockville, MD). The MSD pseudo-neutralization assay was used to measure antibodies neutralizing the binding of the spike protein and receptor-binding domain (RBD) to the ACE2 receptor. Neutralizing antibodies against B.1.1.7, B.1.351, and D614G strains were measured by S-Fuse neutralization assay and expressed as ED50 (effective dose 50%). T-cell responses were characterized as the frequency of PBMC expressing cytokines (IL-2, IL-17a, IFN-γ, TNF-α, IL-13, CD137, and CD154) after stimulation with S or N sequence overlapping peptide pools. IFN-γ ELISpot assay of PBMCs were performed on PBMC stimulated with RBD or N sequence overlapping peptide pools and expressed as spot-forming cell (SFC) per 1.0 × 10⁶ PBMC.

infection, associated or not to the vaccine (group effect) estimated on both the rate of cell infection and the rate of the loss of infected cells. To this aim, we performed a systematic screening by adjusting the model for each marker and we compared these new models with the model without covariates and with the model adjusted for the groups. In particular, our approach allowed us to benefit from all the information provided by the overall dynamics of the immunological markers after the exposure by integrating them as time-varying covariates (see the Materials and methods section for a detailed description of the algorithm). We demonstrate that the RBD-ACE2-binding inhibition measure is sufficient to capture most of the effect of the groups on the infection of target cells (*Figure 4A and B*). The integration of this marker in the model explains the variability of the cell infection rate with greater certainty than the group of intervention, reducing the unexplained variability by 87% compared to 66% (*Supplementary file 1*). The marker actually takes into account the variation between animals within the same group. Hence, it suggests that the levels of anti-RBD antibodies induced by the vaccine that block attachment to ACE2 are highly efficient at reflecting the neutralization of new infections in vivo. Furthermore, when taking into account the information provided by the RBD-ACE2-binding inhibition assay, the effect of the group of intervention was no longer significant (*Supplementary file 1*). Finally, we looked at the estimated viral infectivity according to the binding inhibition assay in each animal. A positive dependence was found between the viral infectivity and the RBD-ACE2-binding inhibition measure, linking an increase of $10^3$ AU of the marker, whether over time or between animals, with an increase of 1.8% (95CI% [1.2%; 2.3%]) of the viral infectivity (see *Supplementary file 4*). Accordingly, the values at the time of exposure were not overlapping at all, distinguishing clearly the vaccinated and unvaccinated animals (see *Figure 4C*).

In the next step, several markers (IgG-binding anti-RBD antibodies, CD8$^+$ T cells producing IFN-γ) appeared to be associated to the rate of loss of infected cells (*Figure 4—figure supplement 1A*). Both specific antibodies and specific CD8$^+$ T cells are mechanisms commonly considered important for killing infected cells. We retained the anti-RBD binding IgG Ab that were positively associated to the increase of the loss of infected cells. For unknown reason the IFN-γ response was high in unstimulated conditions in the naïve group. Thus, although this marker was associated with a decrease of the loss rate of infected cells, it appears essentially here as an indicator of the animal group. Further studies would be needed to fully confirm the place of IFN-γ response as a mechanistic marker.

A large part of the variation of the infection rate (71%) and loss rate of infected cells (60%) were captured by the two markers of CoP: the RBD-ACE2-binding inhibition and the anti-RBD-binding Ab concentration. Using the estimated parameters, the effective reproduction number could be calculated (*R*) which is representing the number of cells secondarily infected by virus from one infected cell (*Figure 4D*). When looking at this effective reproduction number according to the groups, the vaccinated animal presented from the first day of challenge an effective *R* below 1 meaning that no propagation of the infection started within the host. These results were consistent when taking the value of RBD-ACE2-binding inhibition at the time of the challenge without considering the evolution of the inhibition capacity over time (*Figure 4—figure supplement 1B*). This means that the dynamics of the viral replication is impacted very early during the infection process in immunized (i.e., both convalescent and vaccinated) animals and that vaccinated animals were protected from the beginning by the humoral response. Then, we looked at the threshold of the markers of interest leading to the control of the within-host infection (as defined by *R*<1) which was around 30,000 AU for the RBD-ACE2-binding inhibition assay. For the animals in the naïve and the convalescent groups, the observed values of binding inhibition measured by ECL RBD (the lower the better) and of IgG anti-RBD-binding antibodies (the higher the better) led to *R*>1, whereas in vaccinated animals, the value of ECL RBD led to *R*<1. Therefore, our modelling study shows that the inhibition of binding of RBD to ACE2 by antibodies is sufficient to control initial infection of the host (*Figure 4E*). According to the observed value of ECL RBD in vaccinated animals (e.g., 66 AU in *Figure 4E*), a decrease of more than 2 $\log_{10}$ of the inhibition capacity (to reach 81,000 AU), due to variant of concern (VoC) or waning of immunity, would have been necessary to impair the control of the within-host infection. Moreover, a decrease of the neutralizing activity (i.e., increased ECL) could be compensated by an increase of cell death as measured by an increase of binding IgG anti-RBD as a surrogate. As an example, increasing IgG anti-RBD from 2.5 to 10 in the animal MF7 of the convalescent group would lead to a control of the infection.

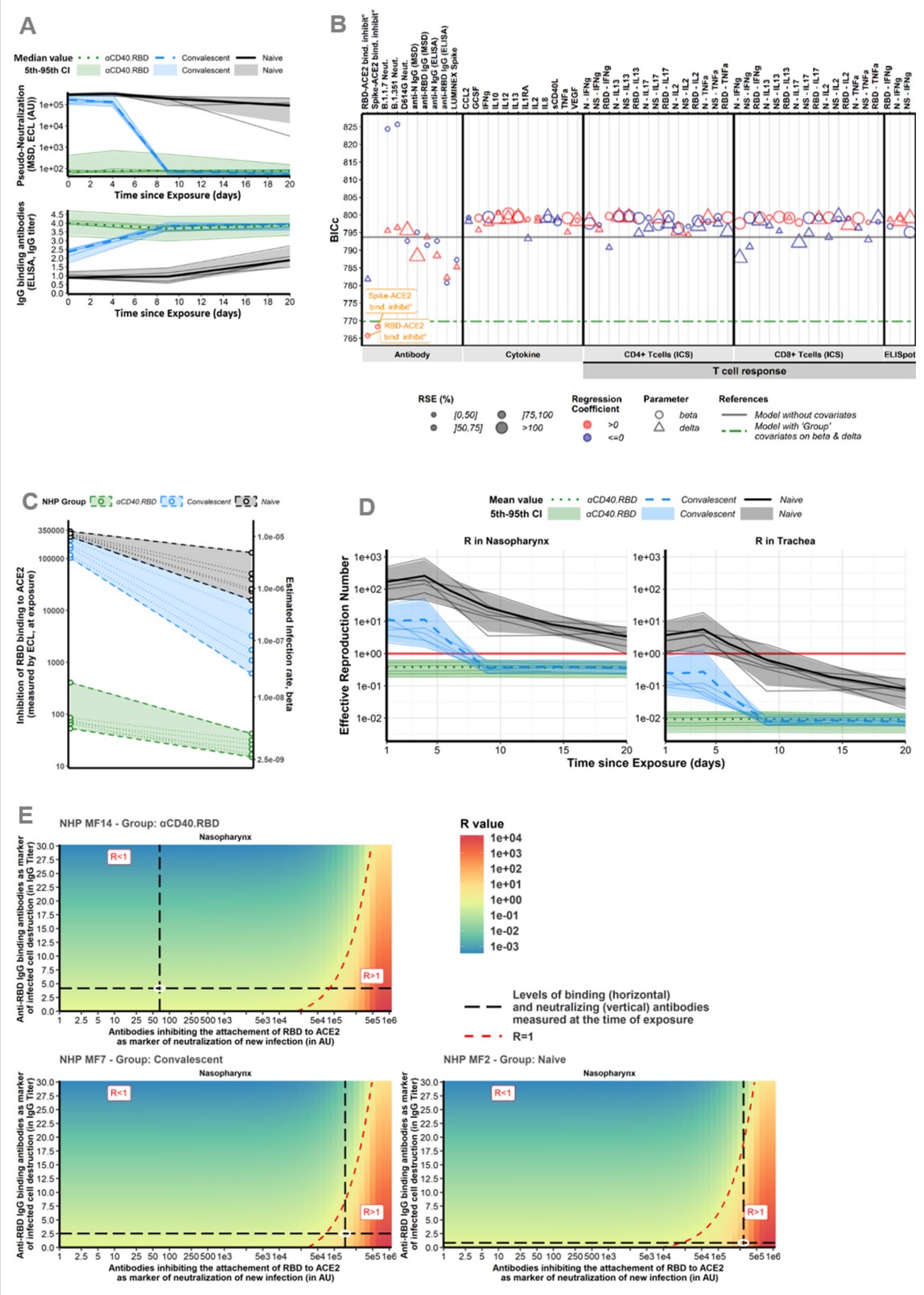

**Figure 4.** Immune markers. (**A**) *Dynamics of biomarker selected as mechanistic correlate of protection (mCoP)*. Quantification of antibodies inhibiting RBD-ACE2 binding, measured by the Mesoscale Discovery (MSD) pseudo-neutralization assay (electro-chemiluminescence [ECL], in arbitrary unit [AU]) (top) and anti-RBD IgG titrated by ELISA assay (in IgG titer) (bottom). Thin lines represent individual values. Thick lines indicate medians of observations within naïve (black, solid line), convalescent (blue, dashed line), and αCD40.RBD-vaccinated convalescent (green, dotted line) animals. Shaded areas

*Figure 4 continued on next page*

*Figure 4 continued*

indicate 5th–95th confidence intervals of observations. (**B**) *Systematic screening of effect of the markers*. For every single marker, a model has been fitted to explore whether it explains the variation of the parameter of interest better or as well than the group indicator. Parameters of interest were $\beta$, the infection rate of ACE2+ target cells, and $\delta$, the loss rate of infected cells. Models were compared according to the Bayesian information criterion (BIC), the lower being the better. The green line represents the reference model that includes the group effect (naïve/convalescent/vaccinated) without any adjustment for immunological marker (see *Figure 3* for more details about measurement of immunological markers). (**C**) *Thresholds of inhibition of RBD-ACE2 binding*. Estimated infection rate (in $(copies/mL)^{-1}$ $day^{-1}$) of target cells according to the quantification of antibodies inhibiting RBD-ACE2 (in ECL) at exposure. Thin dotted lines and circles represent individual values of infection rates (right axis) and neutralizing antibodies (left axis). Shaded areas delimit the pseudo-neutralization/viral infectivity relationships within each group. (**D**) *Reproduction number over time*. Model predictions of the reproduction number over time in the trachea (right) and nasopharynx (left). The reproduction number is representing the number of infected cells from one infected cell if target cells are unlimited. Below one, the effective reproduction number indicates that the infection is going to be cured. Horizontal solid red lines highlight the threshold of one. Same legend than (A). (**E**) *Conditions for controlling the infection*. Basic reproduction number ($R_0$) at the time of the challenge according to the levels of antibodies inhibiting RBD-ACE2 binding (the lower the better) and of anti-RBD IgG-binding antibodies (the higher the better) assuming they are mechanistic correlates of blocking new cell infection and promoting infected cell death, respectively. The red area with $R>1$ describes a situation where the infection is spreading. The green area with $R<1$ describes a situation where the infection is controlled. The dotted red line delimitates the two areas. Black long dashed lines represent the values of neutralizing and binding antibodies measured at exposure. Observed values for three different animals belonging to the naïve (bottom, right), convalescent (bottom, left), and vaccinated (top, left) groups are represented. For each animal, individual values of $R_0$ were estimated considering their individual values of the model parameters ($\beta$ and $\delta$).

The online version of this article includes the following source data and figure supplement(s) for figure 4:

**Source data 1.** Anti-N and anti-receptor-binding domain (RBD)-binding antibodies longitudinally measured post-immunization and quantified by ELISA in the study 1.

**Source data 2.** Anti-receptor-binding domain (RBD) and anti-spike neutralizing antibodies longitudinally measured post-exposure and quantified by Mesoscale Discovery (MSD) assay (in electro-chemiluminescence [ECL]) in the study 1.

**Figure supplement 1.** Immune markers selection and Basic reproduction number.

**Figure supplement 2.** Flowchart of the algorithm for automatic selection of covariate.

In conclusion, the αCD40.RBD vaccine-elicited humoral response leads to the blockade of new cell infection that is well captured by measure of the inhibition of attachment of the virus to ACE2 through the RBD of the spike protein. Hence, the inhibition of binding of RBD to ACE2 is a promising mechanistic CoP. Indeed, this CoP fulfills the three criteria of leading to the best fit (lower BIC), the best explanation of interindividual variability, and fully captured the effect of the group of intervention.

## The model revealed the same CoP related to another protein-based vaccine but not with mRNA-1273 vaccine

We took the opportunity of another study testing a two-component spike nanoparticle protein-based vaccine performed in the same laboratory and using the same immune and virological assays (*Brouwer et al., 2021*), measured only at the time of exposure, for applying the proposed model and methodology. In this study, six animals were vaccinated and compared to four naïve animals (*Figure 5A and B*). The good fit of the data (*Figure 5C and D*) allows for estimating the effect of the vaccine that appeared here also to decrease the infectivity rate (by 99%) and increase the clearance of the infected cells by 79%. Looking at the best mechanistic CoP following the previously described strategy, we ended here again with the inhibition of RBD binding to ACE2 as measured by ECL RBD. In fact, this marker measured at baseline before challenge fulfilled the three criteria: (i) it led to the best model in front of a model adjusted for group effect, (ii) it rendered the group effect non-significant, and (iii) it explained around 71% of the infectivity rate variability, compared to 65% of variability explained by the groups. Interestingly, here again, the inhibition assay led to a clear separation of the estimated rate of infectivity between vaccinees and the placebo group (*Figure 5E*).

Finally, we applied our approach to a published NHP study performed to evaluate several doses of mRNA-1273 vaccine (*Corbett et al., 2020*). Using available data, we compared the viral dynamics in the 100 µg, 10 µg, and placebo groups, enrolling a total of 12 rhesus macaques in a 1:1:1 ratio. Similar to the previous study, only immune markers measured at the time of exposure were available in this study, in addition to viral dynamics. We started from the same model as defined previously. We estimated a reduction of the infection rate by 97% but we did not find any additional effect. Looking at potential mechanistic CoP, we retained neutralization as measured on live cells with Luciferase marker. Although this marker led to the best fit and replaced the group effect (which was non-significant after

Immunology and Inflammation

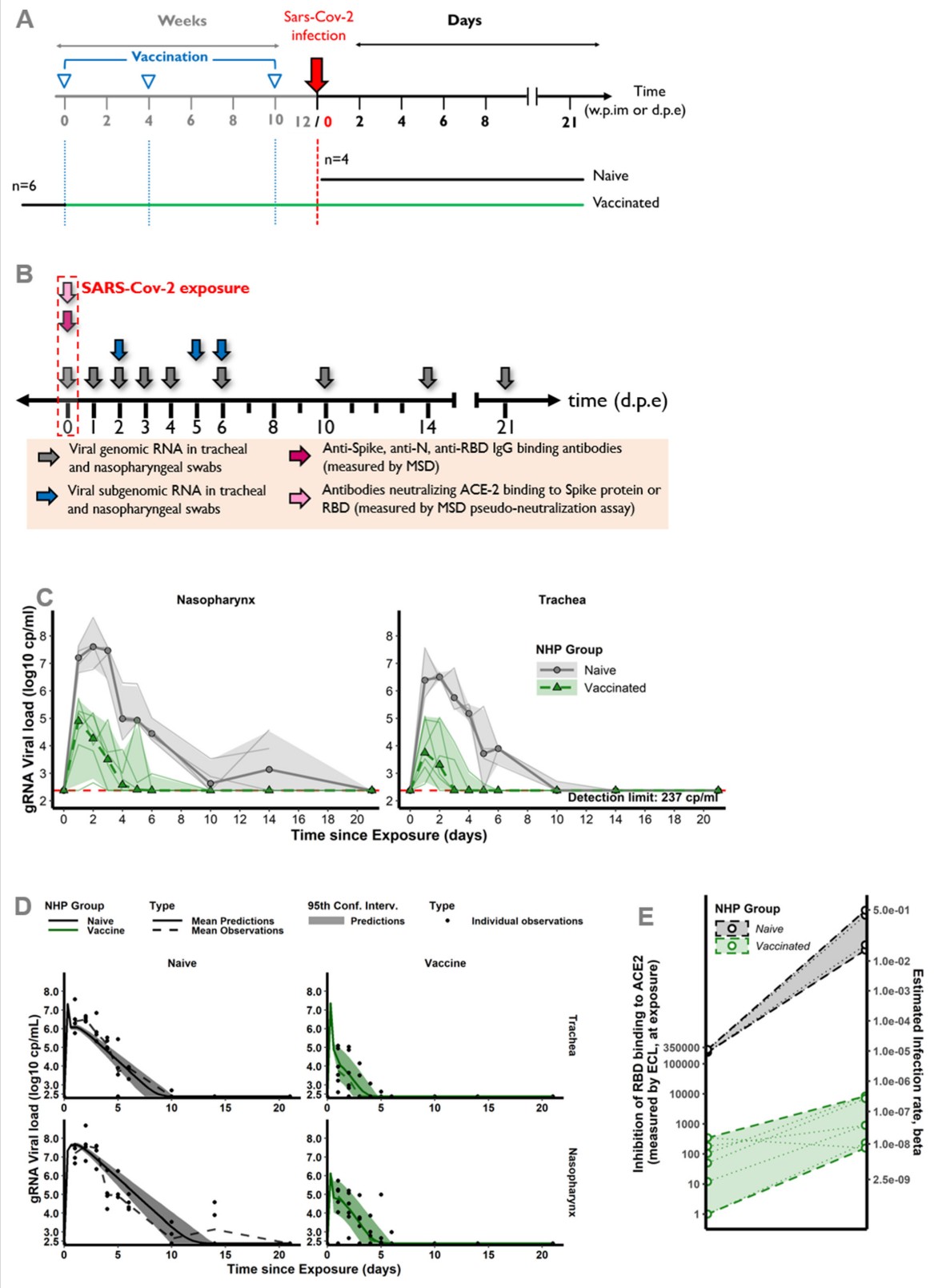

**Figure 5.** Study design and modeling results for the second study testing two-component spike nanoparticle vaccine.
 (**A**) Study design. Cynomolgus macaques were randomly assigned in two experimental groups. Twelve, eight, and two weeks post-infection with SARS-CoV-2 virus, six of them were successively immunized with 50 µg of SARS-CoV-2 S-I53-50NP vaccine. The four other animals received no vaccination.
Two weeks after the final immunization, all monkeys were exposed to a total dose of 10⁶ pfu of SARS-CoV-2 virus via intra-nasal and intra-tracheal routes.

*Figure 5 continued on next page*

*Figure 5 continued*

(**B**) Harvest times and measurements. Nasopharyngeal and tracheal fluids were collected at 0, 1, 2, 3, 4, 5, 6, 10, 14, and 21 days post-exposure (d.p.e.) while blood was taken at 0, 2, 4, 6, 10, 14, and 21 d.p.e. Genomic and subgenomic viral loads were measured by RT-qPCR. Anti-spike, anti-RBD, and anti-nucleocapside (**N**) IgG were titrated using a multiplexed immunoassay developed by Mesoscale Discovery (MSD, Rockville, MD) and expressed in AU mL$^{-1}$. The MSD pseudo-neutralization assay was used to quantify antibodies neutralizing the binding of the spike protein and RBD to the ACE2 receptor and results were expressed in electro-chemiluminescence (ECL). (**C**) Genomic viral load dynamics in nasopharyngeal and tracheal swabs after the exposure to SARS-Cov-2 in naïve (black, solid line) and vaccinated (green, dashed line) animals. Thin lines represent individual values. Thick lines indicate medians within each group. (**D**) Model fit to the log$_{10}$-transformed observed genomic RNA (gRNA) viral load in nasopharynx and trachea after the exposure to SARS-CoV-2 in naïve and vaccinated macaques. Solid thin lines indicate individual dynamics predicted by the model adjusted for groups. Thick dashed lines indicate mean viral load over time. (**E**) Thresholds of inhibition of RBD-ACE2 binding. Estimated infection rate of target cells ((copies/mL)$^{-1}$ day$^{-1}$) according to the quantification of antibodies inhibiting RBD-ACE2 binding (ECL) at exposure for naïve (black) and vaccinated (green) animals. Thin dotted lines and circles represent individual infection rates (right axis) and neutralizing antibodies (left axis). Thick dashed lines and dashed areas delimit the pseudo-neutralization/viral infectivity relationships within each group. (**C,D**) Horizontal red dashed lines represent the limit of quantification and shaded areas the 95% confidence intervals.The second study testing two-component spike nanoparticle vaccine.

The online version of this article includes the following source data for figure 5:

**Source data 1.** Anti-spike, anti-receptor-binding domain (RBD), and anti-N-binding antibodies quantified by Mesoscale Discovery (MSD) assay (AU mL$^{-1}$), and quantification of the spike/ACE2-binding inhibition by MSD assay (in 1/ECL), at the time of exposure in the study 2.

**Source data 2.** Genomic RNA (gRNA) and subgenomic RNA (sgRNA) viral loads longitudinally measured in the trachea and nasopharynx in the study 2.

adjustment for the marker), it explained only 15% of the variability of estimated viral infectivity, while 19% were explained by the groups.

In conclusion, we demonstrated, based upon challenge studies in NHP vaccinated with two different protein-based vaccine platforms, that both vaccines lead to the blockade of new cell infection. Neutralizing antibodies likely represent a consistent mechanistic correlate of protection (mCoP). This could change across vaccine platforms especially because mechanisms of action are different.

## Discussion

We explored the mechanistic effects of three SARS-CoV-2 vaccines and assessed the quality of markers as mCoP. This model showed that neutralizing and binding antibodies elicited by a non-adjuvanted protein-based vaccine targeting the RBD of spike to the CD40 receptor of antigen presenting cells are reliable mCoP. Interestingly, we found the simpler and easier to standardize and implement binding inhibition assay may be more relevant to use as a CoP than cell-culture neutralization assays. This result has been replicated in another study testing a nanoparticle spike vaccine. The model was able to capture the effect of the vaccines on the reduction of the rate of infection of target cells and identified additional effects of vaccines beyond neutralizing antibodies. This latter consisted of increasing the loss rate of infected cells which was better reflected by the IgG-binding antibodies and CD8$^+$ T-cell responses in the case of the CD40-targeting vaccine. One limitation of our study is that the prediction potential of our model relies on the range of the immune markers measured. However, our approach would allow a full exploitation of the data generated as in systems serology where non-neutralizing Ab functions, such as ADCC, ADCP, ADCD, and Ab-dependent respiratory burst (ADRB) are explored (*Chung et al., 2015*). The role of ADCC in natural infection has been previously shown (*Dufloo et al., 2021*), ADCD in DNA vaccine recipients (*Yu et al., 2020*) and with Ad26 vaccine (*Alter et al., 2021*). Here, we extended significantly these data by modelling the viral dynamic, showing that two other protein-based vaccines exert an additional effect on infected cell death which relied on the level of IgG anti-RBD-binding antibodies especially for the CD40.RBD-targeting vaccine. Measurements of other non-neutralizing Ab functions would probably also capture this additional effect.

The next question after determining which marker is a valid mCoP is to define the concentration that leads to protection, looking for a threshold effect that will help to define an objective (*Khoury et al., 2021*; *Jin et al., 2021*). In the context of SARS-CoV-2 virus, several emerged variants are leading to a significant reduction of viral neutralization as measured by various approaches. However, a 20-fold reduction of viral neutralization might not translate in 20-fold reduction of vaccine efficacy (*Emary et al., 2021*). First, there are many steps between viral neutralization and the reduction of viral infectivity or the improvement of clinical symptoms. Second, the consequences of a reduction of viral neutralization could be alleviated by other immunological mechanisms not compromised by

the variant. In the context of natural immunity, when the level of neutralizing antibodies was below a protective threshold, the cellular immune response appeared to be critical (*McMahan et al., 2021*; *Chandrashekar et al., 2020*). We showed with our model that an improvement of infected cell destruction could help to control the within-host infection and is quantitatively feasible.

The control of viral replication is the key for reducing infectivity (*Leung et al., 2020*; *Marks et al., 2021*) as well as disease severity (*Néant, 2021*; *Gutmann et al., 2021*). According to our non-linear model linking the neutralization to the viral replication, a decrease of 4- to 20-fold in neutralization as described for the variants of concern (*Planas et al., 2021*; *Zhou et al., 2021*) is not enough, especially in the context of the response to CD40.RBD-targeting vaccine, to compromise the control of viral replication. The results showing a conserved effectiveness of mRNA vaccines in humans infected by the alpha or beta variants (*Charmet et al., 2021*), although a decrease of neutralization has been reported (*Planas et al., 2021*), are consistent with this hypothesis. However, this is highly dependent upon the mode of action of currently used vaccines and upon the VoC that may much more compromise the neutralization but being also intrinsically less pathogenic such as Omicron (*Nyberg et al., 2022*).

The analysis performed extended significantly the observation of associations between markers as previously reported for SARS-CoV-2 vaccine (*Yu et al., 2020*) and other vaccines (*Kester et al., 2009*) because it allows a more causal interpretation of the effect of immune markers. However, our modelling approach requires the in vivo identification of the biological parameters under specific experimentations. On the other hand, the estimation of parameters included in our model also provided information on some aspect of the virus pathophysiology. Notably, we found an increased capacity of virion production in nasopharynx compared to the trachea which could be explained by the difference in target cells according to the compartment (*Travaglini et al., 2020*). This result needs to be confirmed as it may also be the consequence of a different local immune response (*Pizzorno et al., 2020*). The choice of the structural model defining the host-pathogen interaction is a fundamental step in the presented approach. Here, it was well guided by the biological knowledge, the existing models for viral dynamics (*Goyal et al., 2019*; *Gonçalves et al., 2021*; *Smith et al., 2018*), and the statistical inference allowing the selection of the model that best fit the data. As the number of observations was relatively small in regard to the number of model parameters, we investigated overfitting issues. This was done using a bootstrap approach to evaluate the stability of confidence intervals of the estimated parameters. Results are provided in Appendix 2 'BICc as selection criteria and multiple testing adjustment'. Many modelling choices for the statistical model were made in this approach and more theoretical work evaluating the robustness of the results in their regards may be relevant for future works. In particular, we could relax the constraint of linear interpolation of marker dynamics by using simple regression models, allowing in the same time the integration of error model to account for measurement error for time-varying covariates (*Dafni and Tsiatis, 1998*; *Carroll et al., 2006*; *Wu, 2009*). Moreover, by construction, we assumed similar interindividual variability and effects of covariates within the two URT compartments as well as similar values for the viral infectivity and the loss rate of infected cells. Viral load dynamics measured in lungs being different from those in the URT (*Lui et al., 2020*; *Goyal et al., 2020*), the relaxation of this hypothesis of homogeneous physiological behavior in the URT may be pertinent to extend the model to the LRT. Finally, it should be underlined that the dynamics of the immune response has not been modelled as suggested for instance for B-cell response (*Balelli et al., 2020*). This clearly constitutes the next step after the selection of the markers of interest as done in the present work.

In conclusion, the modelling of the response to two new promising SARS-CoV-2 vaccines in NHP revealed a combination of effects with a blockade of new cell infections and the destruction of infected cells. For these two vaccines, the antibody inhibiting the attachment of RBD to ACE2 appeared to be a very good surrogate of the vaccine effect on the rate of infection of new cells and therefore could be used as a mechanistic CoP. This modelling framework contributes to the improvement of the understanding of the immunological concepts by adding a quantitative evaluation of the contributions of different mechanisms of control of viral infection. In terms of acceleration of vaccine development, our results may help to develop vaccines for 'hard-to-target pathogens', or to predict their efficacy in aging and particular populations (*Pollard and Bijker, 2021*). It should also help in choosing vaccine dose, for instance at early development (*Rhodes et al., 2018*) as well as deciding if and when boosting vaccination is needed in the face of waning protective antibody levels (*Gaebler et al., 2021*;

*Vanshylla et al., 2021*), at least in NHP studies although the framework could be extended to human studies using mixed approaches of within and between hosts modelling (*Goyal et al., 2022*) providing that enough information is collected.

## Materials and methods

### Experimental model and subjects details

Cynomolgus macaques (*Macaca fascicularis*), aged 37–66 months (18 females and 13 males) and originating from Mauritian AAALAC certified breeding centers were used in this study. All animals were housed in IDMIT facilities (CEA, Fontenay-aux-roses), under BSL2 and BSL-3 containment when necessary (Animal facility authorization #D92-032-02, Préfecture des Hauts de Seine, France) and in compliance with European Directive 2010/63/EU, the French regulations and the Standards for Human Care and Use of Laboratory Animals, of the Office for Laboratory Animal Welfare (OLAW, assurance number #A5826-01, US). The protocols were approved by the institutional ethical committee 'Comité d'Ethique en Expérimentation Animale du Commissariat à l'Energie Atomique et aux Energies Alternatives' (CEtEA #44) under statement number A20-011. The study was authorized by the 'Research, Innovation and Education Ministry' under registration number APAFIS#24434-2020030216532863v1.

### Evaluation of anti-spike, anti-RBD, and neutralizing IgG antibodies

*Anti-spike IgG were titrated by multiplex bead assay.* Briefly, Luminex beads were coupled to the spike protein as previously described (*Fenwick et al., 2021*) and added to a Bio-Plex plate (Bio-Rad). Beads were washed with PBS 0.05% tween using a magnetic plate washer (MAG2x program) and incubated for 1 hr with serial diluted individual serum. Beads were then washed and anti-NHP IgG-PE secondary antibody (Southern Biotech, clone SB108a) was added at a 1:500 dilution for 45 min at room temperature (RT). After washing, beads were resuspended in a reading buffer 5 min under agitation (800 rpm) on the plate shaker then read directly on a Luminex Bioplex 200 plate reader (Bio-Rad). Average MFI from the baseline samples were used as reference value for the negative control. Amount of anti-spike IgG was reported as the MFI signal divided by the mean signal for the negative controls.

*Anti-RBD and anti-nucleocapside (N) IgG* were titrated using a commercially available multiplexed immunoassay developed by Mesoscale Discovery (MSD, Rockville, MD) as previously described (*Johnson et al., 2020*). Briefly, antigens were spotted at 200–400 µg mL$^{-1}$ in a proprietary buffer, washed, dried, and packaged for further use (MSD Coronavirus Plate 2). Then, plates were blocked with MSD Blocker A following which reference standard, controls, and samples diluted 1:500 and 1:5000 in diluent buffer were added. After incubation, detection antibody was added (MSD SULFO-TAGTM Anti-Human IgG Antibody) and then MSD GOLDTM Read Buffer B was added and plates read using a MESO QuickPlex SQ 120 MM Reader. Results were expressed as arbitrary unit (AU) mL$^{-1}$.

*Anti-RBD and anti-N IgG* were titrated by ELISA. The nucleocapsid and the spike RBD (Genbank # NC_045512.2) were cloned and produced in *Escherichia coli* and CHO cells, respectively, as previously described (*Flamar et al., 2012*). Antigens were purified on C-tag column (Thermo Fisher) and quality-controlled by SDS-PAGE and for their level of endotoxin. Antigens were coated in a 96-well plates Nunc-immuno Maxisorp (Thermo Fisher) at 1 µg mL$^{-1}$ in carbonate buffer at 4°C overnight. Plates were washed in TBS Tween 0.05% (Thermo Fisher) and blocked with PBS 3% BSA for 2 hr at RT. Samples were then added, in duplicate, in serial dilution for 1 hr at RT. Non-infected NHP sera were used as negative controls. After washing, anti-NHP IgG coupled with HRP (Thermo Fisher) was added at 1:20,000 for 45 min at RT. After washing, TMB substrate (Thermo Fisher) was added for 15 min at RT and the reaction was stopped with 1 M sulfuric acid. Absorbance of each well was measured at 450 nm (reference 570 nm) using a Tristar2 reader (Berthold Technologies). The EC50 value of each sample was determined using GraphPad Prism 8 and antibody titer was calculated as log (1/EC50).

*The MSD pseudo-neutralization assay* was used to measure antibodies neutralizing the binding of the spike protein to the ACE2 receptor. Plates were blocked and washed as above, assay calibrator (COVID- 19 neutralizing antibody; monoclonal antibody against S protein; 200 µg mL$^{-1}$), control sera, and test sera samples diluted 1:10 and 1:100 in assay diluent were added to the plates. Following incubation of the plates, an 0.25 µg mL$^{-1}$ solution of MSD SULFO-TAGTM-conjugated ACE2 was added after which plates were read as above. Electro-chemiluminescence (ECL) signal was recorded.

## Viral dynamics modelling

The mechanistic approach we developed to characterize the impact of the immune response on the viral gRNA and sgRNA dynamics relies on a mechanistic model divided in three layers: first, we used a mathematical model based on ODEs to describe the dynamics in the two compartments, the nasopharynx and the trachea. Then, we used a statistical model to take into account both the interindividual variability and the effects of covariates on parameters. Finally, we considered an observation model to describe the observed $\log_{10}$ viral loads in the two compartments.

For the mathematical model, we started from previously published models (*Gonçalves et al., 2020*; *Kim et al., 2021*; *Baccam et al., 2006*) where the nasopharynx and trachea were respectively described by a target cell limited model, with an eclipse phase, as model of acute viral infection assuming target-cell limitation (*Baccam et al., 2006*). We completed the model by adding a compartment for the inoculum that distinguishes the injected virus ($V_s$) from the virus produced de novo ($V_i$ and $V_{ni}$). To our knowledge, this distinction has not been proposed in any previous work. Two main reasons led us to make this choice. First, it allowed us to study the dynamics of the inoculum, in particular during the early phase of viral RNA load dynamics. Second, as described in more detail below, it gave us the opportunity to use all the information provided by the preclinical studies, such as the known number of inoculated virions, to define the initial conditions of the ODE model rather than estimating or randomly fixing them for $V_i$ and $V_{ni}$, as is usually done. Consequently, for each of the two compartments, the model included uninfected target cells ($T$) that can be infected ($I_1$) either by infectious viruses ($V_i$) or inoculum ($V_s$) at an infection rate $\beta$. After an eclipse phase, infected cells become productively infected cells ($I_2$) and can produce virions at rate $P$ and be lost at a per capita rate $\delta$. The virions generated can be infectious ($V_i$) with proportion $\mu$ while the $(1-\mu)$ remaining proportion of virions is non-infectious ($V_{ni}$). Mathematically, a single compartment ($V$) for de novo produced virions could be considered in the model, with $\mu V$ and $(1-\mu)V$ representing the respective contributions of infectious and non-infectious viruses to the biological mechanisms. However, to have a better visual understanding of the distinction between the two types of viruses, we wrote the model with distinct compartments, $V_i$ and $V_{ni}$.

Finally, virions produced de novo and those from the inoculum are cleared at a rate $c$ and $c_i$, respectively. Distinct clearances were considered to account for the effects of experimental conditions on viral dynamics. In particular, it is hypothesized that, animals being locally infected with large numbers of virions, a large proportion of it is assumed to be rapidly eliminated by swallowing and natural downstream influx, in contrast to the de novo-produced virions. However, it is important to keep in mind that this distinction was possible because of the controlled experimental conditions performed in animals, (i.e., exact timing and amount of inoculated virus known, and frequent monitoring during the early phase of the viral dynamics). Because of identifiability issues, similar clearances for infectious and non-infectious viruses were used. Accordingly, the model can be written as the following set of differential equations, where the superscript $X$ denotes the compartment of interest ($N$, nasopharynx or $T$, trachea):

$$
\begin{cases}
\frac{dT^X}{dt} = -\beta^X V_i^X T^X - \mu \beta^X V_s^X T^X \\
\frac{dI_1^X}{dt} = \beta^X V_i^X T^X + \mu \beta^X V_s^X T^X - k I_1^X \\
\frac{dI_2^X}{dt} = k I_1^X - \delta^X I_2^X \\
\frac{dV_i^X}{dt} = \mu P^X I_2^X - c V_i^X - \beta^X V_i^X T^X \\
\frac{dV_{ni}^X}{dt} = (1 - \mu) P^X I_2^X - c V_{ni}^X \\
\frac{dV_s^X}{dt} = -c_i V_s^X - \mu \beta^X V_s^X T^X \\
\\
T^X (t = 0) = T_0^X \,; I_1^X (t = 0) = 0 \,; I_2^X (t = 0) = 0 \\
V_i^X (t = 0) = 0 \,; V_{ni}^X (t = 0) = 0 \,; V_s^X (t = 0) = V_{S,0}^X
\end{cases}
\tag{1}
$$

where $T^X (t = 0)$, $I_1^X (t = 0)$, $I_2^X (t = 0)$, $V_i^X (t = 0)$, $V_{ni}^X (t = 0)$, and $V_s^X (t = 0)$ are the initial conditions at the time of exposure. The initial concentration of target cells, that are the epithelial cells expressing the ACE2 receptor, is expressed as $T_0^X = \frac{T_0^{X,nbc}}{W^X}$, where $T_0^{X,nbc}$ is the initial number of cells and $W^X$ is

the volume of distribution of the compartment of interest (see the subsection 'Consideration of the volume of distribution'). Each animal was exposed to $1 \times 10^6$ pfu of SARS-CoV-2 representing a total of $2.19 \times 10^{10}$ virions. Over the total inoculum injected (5 mL), 10% (0.5 mL) and 90% (4.5 mL) of virions were respectively injected by the intra-nasal route and the intra-tracheal route leading to the following initial concentrations of the inoculum within each compartment: $V_{S,0}^N = \frac{0.10 \times Inoc_0}{W^N}$ and $V_{S,0}^T = \frac{0.90 \times Inoc_0}{W^T}$, with $Inoc_0$ the number of virions injected via the inoculum.

Using the gRNA and sgRNA viral loads, we estimated the viral infectivity, the viral production rate, and the loss rate of infected cells within each of the two compartments of the URT (**Supplementary file 2**). To account for interindividual variability and covariates, each of those three parameters was described by a mixed-effect model and jointly estimated between the two compartments as follows:

$$
\begin{cases}
\log_{10}\left(\beta_i^N\right) = \beta_0 + \phi_{conv}^\beta \times I_{group=conv} + \phi_{CD40}^\beta \times I_{group=CD40} + u_i^\beta \\
\beta_i^T = \beta_i^N \times \exp\left(f_\beta^T\right) \\
\log\left(\delta_i^N\right) = \log\left(\delta_0\right) + \phi_{conv}^\delta \times I_{group=conv} + \phi_{CD40}^\delta \times I_{group=CD40} + u_i^\delta \\
\delta_i^T = \delta_i^N \times \exp\left(f_\delta^T\right) \\
\log\left(P_i^N\right) = \log\left(P_0\right) + \phi_{conv}^P \times I_{group=conv} + \phi_{CD40}^P \times I_{group=CD40} + u_i^P \\
P_i^T = P_i^N \times \exp\left(f_P^T\right)
\end{cases}
\tag{2}
$$

where $\beta_0$, $\log\left(\delta_0\right)$, and $\log\left(P_0\right)$ are the fixed effects, $\left\{\phi_{conv}^\theta \mid \theta \in \{\beta, \delta, P\}\right\}$ and $\left\{\phi_{CD40}^\theta \mid \theta \in \{\beta, \delta, P\}\right\}$ are respectively the regression coefficients related to the effects of the group of convalescent and αCD40.RBD-vaccinated animals for the parameters $\beta$, $\delta$, and $P$, and $u_i^\theta$ is the individual random effect for the parameter $\theta$, which is assumed to be normally distributed with variance $\omega_\theta^2$. A log-transformation was adopted for the parameters $\delta$ and $P$ to ensure their positivity while a $\log_{10}$-transformation was chosen for viral infectivity to also improve the convergence of the estimation. Because of the scale difference between the parameter $\beta$ and the other parameters (see **Supplementary file 2**), the mere use of the log-transformation for this parameter led to convergence issues. The use of a $\log_{10}$-transformation allowed to overcome this problem. Moreover, as shown in **Equation 2**, a joint estimation of the parameters $\beta$, $\delta$, and $P$ between the two compartments of the URT was considered. In this regard, a homogeneous interindividual variability within the URT was assumed as well as a similar contribution of the covariates to the value of the parameters. Parameters in the trachea were then either equal or proportional to those in the nasopharynx. This modelling choice, resulting in a smaller number of parameters to be estimated, was made mainly to address identifiability issues and to increase the power of the estimation. All other parameters included in the target-cell limited models were assumed to be fixed (see the subsection 'Parameter estimation' for more details).

In practice, after the selection of the optimal statistical model (see Appendix 1 'Model building'), random effects were added only to the parameters $\beta$ and $\delta$ (i.e., $\omega_\beta \neq 0$, $\omega_\delta \neq 0$, and $\omega_P = 0$), and the estimation of multiple models identified the viral production rate $P$ as the only parameter taking different values between the trachea and nasopharynx. (i.e., $\beta^N = \beta^T$ with $f_\beta^T = 0$, $\delta^N = \delta^T$ with $f_\delta^T = 0$, while $P^N \neq P^T$). Finally, the adjustment of the model for the categorical covariates of groups of treatment, natural infection, and/or vaccination identified $\beta$ and $\delta$ as the parameters with a statistically significant effect of these covariates (i.e., $\phi_{conv}^P = 0$ and $\phi_{CD40}^P = 0$).

For the observation model, we jointly described genomic and subgenomic viral loads in the two compartments of the URT. We defined genomic viral load, which characterizes the total viral load observed in a compartment (nasopharynx or trachea), as the sum of inoculated virions ($V_s$), infectious ($V_i$), and non-infectious virions ($V_{ni}$). The sgRNA was described as proportional to the infected cells ($I_1 + I_2$). This choice was driven by two main reasons. First, sgRNA is only transcribed in infected cells (**Sawicki et al., 2007**). Second, as described by **Miao et al., 2011**, to overcome identifiability issues between the parameters $\beta$ and $P$ typically observed in target-cell limited models. The comparison of the two observation models describing sgRNA as either proportional to virions produced de novo ($V_i + V_{ni}$) or proportional to infected cells ($I_1 + I_2$) confirmed this conclusion. In addition to a better BICc value (–25 points) compared with the first model, the second one allowed the estimation of both $\beta$ and $P$ by counteracting identifiability problems faced with the first model (results not shown). Accordingly, the $\log_{10}$-transformed gRNA and sgRNA of the $i$th animal at the $j$th time point in compartment $X$

(nasopharynx or trachea), denoted $gRNA_{ij}^X$ and $sgRNA_{ij}^X$, respectively, were described by the following equations:

$$\begin{cases} gRNA_{ij}^X = \log_{10}\left[\left(V_i^X + V_{ni}^X + V_s^X\right)\left(\Theta_i^X, t_{ij}\right)\right] + \varepsilon_{ij,g}^X \quad \varepsilon_{ij,g}^X \sim N\left(0, \sigma_{gX}^2\right) \\ sgRNA_{ij}^X = \alpha_{sgRNA} \times \log_{10}\left[\left(I_1^X + I_2^X\right)\left(\Theta_i^X, t_{ij}\right)\right] + \varepsilon_{ij,sg}^X \quad \varepsilon_{ij,sg}^X \sim N\left(0, \sigma_{sgX}^2\right) \end{cases} \tag{3}$$

where $\Theta_i^X$ is the set of parameters of the subject $i$ for the compartment $X$ and $\varepsilon$ are the additive normally distributed measurement errors.

## Consideration of the volume of distribution

To define the concentration of inoculum within each compartment after injection, nasopharyngeal and tracheal volumes of distribution, labelled $W^N$ and $W^T$, respectively, were needed. Given the estimated volumes of the trachea and the nasal cavities in four monkeys similar to our 18 macaques (*Figure 2— figure supplement 2A–C*) and the well-documented relationship between the volume of respiratory tract and animal weights (*Asgharian et al., 2012*), the volume of distribution of each compartment was defined as a step function of NHP weights:

$$W_i^N = \begin{cases} 4 & \text{if weight}_i \leq 4.5 \\ 5.5 & \text{otherwise} \end{cases}$$
$$W_i^N = \begin{cases} 2 & \text{if weight}_i \leq 4.5 \\ 3 & \text{otherwise} \end{cases} \tag{4}$$

where weight$_i$ is the weight of the monkey $i$ in kg. Using *Equation 4* and weights of our 18 NHPs (mean = 4.08; [Q1; Q3] = [3.26; 4.77]), we estimated WT = 2 and WN = 4 mL for a third of them (n=12) (*Figure 2—figure supplement 2D*), leading to the initial concentration of target cells $T_0^X$ (see 'Viral dynamics modelling' for equation) fixed at $3.13 \times 10^4$ cells mL$^{-1}$ and $1.13 \times 10^4$ cells mL$^{-1}$ in nasopharynx and trachea, respectively. Similarly, their initial concentrations of challenge inoculum $V_{S,0}^X$ were fixed at $5.48 \times 10^8$ copies,mL$^{-1}$ and $9.86 \times 10^9$ copies,mL$^{-1}$ in nasopharynx and trachea respectively. For the last third of NHPs (n=6), WT = 3 and WN = 5.5 mL leading to $T_0^X$ fixed at $2.27 \times 10^4$ cells mL$^{-1}$ in nasopharynx and $7.50 \times 10^3$ cells mL$^{-1}$ in trachea while $V_{S,0}^X$ was fixed at $3.98 \times 10^8$ copies mL$^{-1}$ in nasopharynx and $6.57 \times 10^9$ copies mL$^{-1}$ in trachea. Through this modelling, we assumed a homogenous distribution of injected virions and target cells within nasopharyngeal and tracheal compartments. In addition, the natural downward flow of inoculum toward lungs, at the moment of injection, was indirectly taken into account by the parameter of inoculum clearance, $c_i$.

## Parameter estimation

Among all parameters involved in the three layers of the mechanistic model, some of them have been fixed based on experimental settings and/or literature. That is the case of the proportion of infectious virus ($\mu$) that has been fixed at 1/1000 according to previous work (*Gonçalves et al., 2021*) and additional work (results not shown) evaluating the stability of the model estimation according to the value of this parameter. The initial number of target cells, that are the epithelial cells expressing the ACE2 receptor, $T_0^{X,nbc}$ was fixed at $1.25 \times 10^5$ cells in the nasopharynx and $2.25 \times 10^4$ cells in trachea (*Gonçalves et al., 2021*; *Supplementary file 2*). The duration of the eclipse phase (1/$k$), the clearance of the inoculum ($c_i$) and the clearance of the virus produced de novo ($c$) were estimated by profile likelihood. The profile likelihood consists in defining a grid of values for the parameters to be evaluated and sequentially fixing these parameters to one of these combinations of values. The model and all the parameters that are not fixed are then estimated by maximizing the log-likelihood. In this process, all parameters that are assumed to be fixed in the model (i.e., $\mu$ and the initial conditions) are held fixed. Finally, the optimal set of parameters is chosen as the one optimizing the log-likelihood. Although the available data did not allow the direct estimation of these three parameters, the use profile likelihood enabled the exploration of various potential values for $k$, $c$, and $c_i$ . In a first step, we explored the 18 models resulting from the combination of three values of $k \in \{1, 3, 6\}$ day$^{-1}$ and six values for $c \in \{1, 5, 10, 15, 20, 30\}$ day$^{-1}$, assuming that the two parameters of virus clearance were

equal, as first approximation. As shown in **Supplementary file 3a**, an eclipse phase of 8 hr ($k$=3) and virus clearance higher than 15 virions per day led to lowest values of –2log-likelihood (–2LL, the lower the better). In a second step, we fixed the parameter $k$ at 3 day$^{-1}$ and estimated the 70 models resulting from the combination of 10 values for $c \in$ {1, 2, 3, 4, 5, 10, 15, 20, 25, 30} day$^{-1}$ and 7 values for $\in$ {1, 5, 10, 15, 20, 25, 30} day$^{-1}$ (**Supplementary file 3b**). The distinction of the two parameters of free virus clearance enabled to find much lower half-life of inoculum (~50 min) than half-life of virus produced de novo (~5.55 hr), with $c$=3 day$^{-1}$ compared to $c_i$ = 20 day$^{-1}$.

Once all these parameters have been fixed, the estimation problem was restricted to the determination of the viral infectivity $\beta$, the viral production rate $P$, the loss rate of infected cells $\delta$ for each compartment, the parameter $\alpha_{vlsg}$ in the observation model, regression coefficients for groups of intervention ($\phi_{conv}, \phi_{CD40}$), and standard deviations for both random effects ($\omega$) and error model ($\sigma$). The estimation was performed by maximum likelihood estimation using a stochastic approximation EM algorithm implemented in the software Monolix (http://www.lixoft.com). The Fisher information matrix was calculated by stochastic approximation, providing for each estimated parameter its variance, from which we were able to derive its 95% confidence interval. Selection of the compartment effect on parameters ($\beta$, $\delta$, $P$) as well as random effects and covariates on the statistical model (**Equation 2**) was performed by the estimation of several models that were successively compared according to the corrected Bayesian information criterion (BICc) (to be minimized). After the removal of random effect on the viral production ($\omega_P = 0$) allowing the reduction of the variance on the two other random effects, all combinations of compartment effects were evaluated, leading to the final selection of a single effect on $P \left( f_\beta^T = f_\delta^T = 0 \right)$. Then, the effect of group intervention was independently added on model parameters among $\beta$, $\delta$, $P$, and $c$. Once the group effect on the viral infectivity identified as the best one, the addition of a second effect on the remaining parameters was tested, resulting in the selection of the loss rate of infected cells. Finally, the irrelevance of the addition of a third effect was verified.

The possibility of migration of free plasma virus between the nasopharynx and the trachea was tested. However, as widely described in the literature, the transport of viral particles within the respiratory tract is negligible in the viral dynamics and is difficult to estimate. The reader can refer to Appendix 1 'Model building' for an additional modelling work conducted to estimate this exchange and provided the same conclusion. Accordingly, the two compartments of the URT were assumed are distinct in our model.

## Algorithm for automatic selection of biomarkers as CoP

After identifying the effect of the group of intervention on both the viral infectivity ($\beta$) and the loss rate of infected cells ($\delta$), we aimed at determining whether some immunological markers quantified in the study could capture this effect. Nowadays, many methods for selecting constant covariates already exist (**Chowdhury and Turin, 2020**) and are implemented in software like Monolix. However, these latter do not allow time-varying covariates. In this section, we present the algorithm we implemented to select time-varying covariates. We proposed a classical stepwise data-driven automatic covariate modelling method (**Figure 4—figure supplement 2**). However, initially implemented to select covariates from more than 50 biomarkers, computational time restricted us to consider only a forward selection procedure. Nevertheless, the method can be easily extended to classical stepwise selection in which both forward selection and backward elimination are performed sequentially. Although the method was developed for time-varying covariates, it can also be applied to constant covariates.

At the initialization step ($k$=0) (see **Figure 4—figure supplement 2**), the algorithm requests three inputs: (**World Health Organization, 2021**) a set of potential $M$ covariates, labelled *Marker m* for $m \in \{1, \cdots, M\}$ (e.g., immunological markers); (**Cobey et al., 2021**) a set of $P$ parameters on which covariates could be added, labelled $\theta_p$ for $p \in \{1, \cdots, P\}$ (e.g., $\beta$ and $\delta$); and (**Kuzmina et al., 2021**) an initial model (e.g., the model without covariates), labelled $M^0$, with $\theta_p^0$ being the definition of the parameter $\theta_p$. At each step $k$>0, we note $M^{k-1}$ the current model resulting in the model built in the step $k$−1. Then, each combination of markers and parameters that have not already been added in $M^{k-1}$, labelled $r \left( r \in \left\{ Marker\, m \bigotimes \theta_p \notin M^{k-1} \mid m \in \{1, \ldots M\}, p \in \{1, \ldots P\} \right\} \right)$, are considered and tested in an univariate manner (each relation $r$ is independently added in $M^{k-1}$ and ran). To this end, the parameter $\theta_p$ involved in this relationship $r$ is modified as $\theta_p^k (t) = \theta_p^{k-1} (t) \times exp \left( \phi_m^p \times Marker_m (t) \right)$, where $\phi_m^p$ is the regression coefficient related to the marker and $Marker_m (t)$ being the trajectory of

the marker over time, while other parameters remain unchanged $\left(\forall \theta_q \notin r, \theta_q^k(t) = \theta_q^{k-1}(t)\right)$. Once all these models evaluated, the one with the optimal value of a given selection criterion defining the quality of the fits (e.g., the lowest BICc value) is selected and compared to the model $M^{k-1}$. If the value of the criterion is better than the one found for $M^{k-1}$, then this model is defined as the new current model, $M^k$, and the algorithm moves to the step $k+1$. Otherwise, the algorithm stops. The algorithm can also be stopped at the end of a fixed number of step $k$.

The objective of this algorithm being to identify mechanistic CoP, at each step, the selected model should respect, in addition to the best fits criterion, the two other criteria defining mCoP meaning the ability to capture the effect of the group of intervention and the ability to better explain the variability on individual parameters than the model adjusted for the group effect. To this end, we verify that in the selected model additionally adjusted for the group of intervention, the group effect appears as non-significantly different from 0 using a Wald test. Then, we check that the variances of random effects in the selected model are lower or equal to the ones obtained in the model adjusted only for the group effect.

## Modelling hypothesis for time-dependent covariates in our application

Using a population-based approach to estimate our mechanistic model and similar to the adjustment of the model for constant covariates (e.g., groups of intervention), time-varying covariates are incorporated into the statistical model as individual-specific explanatory variables in the mixed-effects models. To implement the algorithm for selecting the time-varying covariates, many modelling choices were made. First, targeting covariates able to fully replace the group of intervention, we kept a similar mathematical relationship between parameters and immune markers than the one used with the constant covariate (see *Equation 2*). Accordingly, we adjusted the model parameters additively in logarithmic scale. In this regard, at each step $k$ ($k>0$), the parameter $\theta_p$ was defined as $\log\left(\theta_p^k(t)\right) = \log\left(\theta_p^{k-1}(t)\right) + \phi_m^p \times Marker_m(t)$. However, this choice may affect the results and other choices may be more relevant under different conditions. Second, because immune markers are observed only at discrete time points, whereas the estimation of the model is performed in a continuous way, we introduced immune markers as time-varying covariates using linear interpolation. Lets denote $Marker_{i,j}$ the value of the marker observed for the $i$th animal at the $j$th time point, with $i \in \{1, \ldots, n\}$ and $j \in \{1, \ldots, J\}$. By linear interpolation, the time-continuous marker was defined as, $\forall t > 0$,

$$Marker_i^{int}(t) = \sum_{j=1}^{J-1} I_{[t_j:t_{j+1}]}(t) \left[ \frac{Marker_{i,j+1}-Marker_{i,j}}{t_{j+1}-t_j}t + \frac{Marker_{i,j}t_{j+1}-Marker_{j+1}t_j}{t_{j+1}-t_j} \right] + I_{t \geq t_J}(t) \times Marker_{i,J}$$

As previously described in the Results section, three different studies were considered in this work: a main study reported by *Marlin et al., 2021*, testing the αCD40.RBD vaccine, and two additional studies (*Corbett et al., 2020*; *Brouwer et al., 2021*) evaluating a two-component spike nanoparticle vaccine and the mRN-1273 vaccine, respectively. In the main study, the method was applied with both time-varying covariates and constant covariates for which only baseline value was considered, such that $Marker_i(t)=Marker_i(t=0)$ (see *Supplementary file 1*). For the other two studies, only the baseline values were considered as covariates, the dynamics being not available. To assess the robustness of the results, several selection criteria were tested: AIC, BIC, log-likelihood, the percentage of explained interindividual variability, and similar results were obtained for all (results not shown). Moreover, as presented in Appendix 2 'BICc as selection criteria and multiple testing adjustment', we verified the robustness of the use of BIC as selection criteria despite the multiplicity of the tests. The identification of antibodies inhibiting the attachment of the RBD to the ACE2 receptor (ECLRBD) as the first time-varying CoP led to the definition of the time-varying viral infectivity for the $i$th animal as described in *Equation 5*, while the selection anti-RBD IgG-binding antibodies led to the elimination rate of infected cells given in *Equation 6*.

$$\beta_i(t) = 10^{\beta_0 + u_i^\beta} \times exp\left(\phi_{ecl}^\beta \times ECLRBD_i^{int}(t)\right) \tag{5}$$

$$\delta_i(t) = \delta_0 \times \exp\left(\phi_{igg}^\delta \times IggRBD_i^{int}(t) + u_i^\delta\right) \tag{6}$$

## Quantification and statistical analysis

In each of the three studies used in this work, no statistical tests were performed on the raw data (i.e, observations), whether for viral load or for immune marker measurements, to identify statistical differences between treatment groups, as the statistical analyses were already been performed in the respective papers. Statistical significance of the effect of groups in model estimation is indicated in the tables by stars: *, $p < 0.05$; **, $p < 0.01$; ***, $p < 0.001$ and were estimated by Wald tests (Monolix software version 2019R1).

Model parameters were estimated with the SAEM algorithm (Monolix software version 2019R1). Graphics were generated using R version 3.6.1 and Excel 2016 and details on the statistical analysis for the experiments can be found in the accompanying figure legends. Horizontal red dashed lines on graphs indicate assay limit of detection.

## Acknowledgements

We would like to thank J Guedj and O Terrier for fruitful discussions on the model definition. We thank S Langlois, J Demilly, N Dhooge, P Le Calvez, M Potier, JM Robert, T Prot, and C Dodan for the NHP experiments; L Bossevot, M Leonec, L Moenne-Loccoz, M Calpin-Lebreau, and J Morin for the RT-qPCR, ELISpot and Luminex assays, and for the preparation of reagents; A-S Gallouët, M Gomez-Pacheco, and W Gros for NHP T-cell assays and flow cytometry; B Fert for her help with the CT scans; M Barendji, J Dinh, and E Guyon for the NHP sample processing; S Keyser for the transports organization; F Ducancel and Y Gorin for their help with the logistics and safety management; I Mangeot for her help with resources management and B Targat contributed to data management. The monkey and syringe pictures in *Figure 1* was created with BioRender.com. This work was supported by INSERM and the Investissements d'Avenir program, Vaccine Research Institute (VRI), managed by the ANR under reference ANR-10-LABX-77-01. MA has been funded by INRIA PhD grant. The Infectious Disease Models and Innovative Therapies (IDMIT) research infrastructure is supported by the 'Programme Investissements d'Avenir', managed by the ANR under reference ANR-11-INBS-0008. The Fondation Bettencourt Schueller and the Region Ile-de-France contributed to the implementation of IDMIT's facilities and imaging technologies used to define volume of respiratory tract. The NHP study received financial support from REACTing, the Fondation pour la Recherche Medicale (FRM; AM-CoV-Path). We thank Lixoft SAS for their support. Numerical computations were in part carried out using the PlaFRIM experimental testbed, supported by Inria, CNRS (LABRI and IMB), Université de Bordeaux, Bordeaux INP, and Conseil Régional d'Aquitaine (see https://www.plafrim.fr). We thank Miles Davenport and Frederik Graw as Senior Editor and Reviewing Editor of our paper, respectively, and the three anonymous reviewers for their time and their constructive comments.

## Additional information

### Funding

| Funder | Grant reference number | Author |
| --- | --- | --- |
| Agence Nationale de la Recherche | ANR-10-LABX-77-01 | Yves Levy Rodolphe Thiébaut |
| Agence Nationale de la Recherche | ANR-11- 1018 INBS-0008 | Roger Le Grand |

The funders had no role in study design, data collection and interpretation, or the decision to submit the work for publication.

### Author contributions

Marie Alexandre, Conceptualization, Methodology, Resources, Software, Validation, Visualization, Writing – original draft, Writing – review and editing; Romain Marlin, Conceptualization, Investigation, Resources, Visualization, Writing – original draft, Writing – review and editing; Mélanie Prague, Methodology, Software, Supervision, Validation, Visualization, Writing – original draft, Writing – review and editing; Severin Coleon, Sylvain Cardinaud, Benoit Delache, Mathieu Surenaud, Mathilde Galhaut,

Nathalie Dereuddre-Bosquet, Mariangela Cavarelli, Pauline Maisonnasse, Mireille Centlivre, Christine Lacabaratz, Aurelie Wiedemann, Sandra Zurawski, Gerard Zurawski, Investigation, Resources, Writing – review and editing; Nidhal Kahlaoui, Thibaut Naninck, Investigation, Resources, Visualization, Writing – review and editing; Olivier Schwartz, Rogier W Sanders, Resources, Writing – review and editing; Roger Le Grand, Yves Levy, Conceptualization, Funding acquisition, Project administration, Resources, Supervision, Writing – original draft, Writing – review and editing; Rodolphe Thiébaut, Conceptualization, Funding acquisition, Methodology, Project administration, Supervision, Validation, Writing – original draft, Writing – review and editing

### Author ORCIDs
Marie Alexandre  http://orcid.org/0000-0002-3557-7075
Mélanie Prague  http://orcid.org/0000-0001-9809-7848
Pauline Maisonnasse  http://orcid.org/0000-0002-0555-207X
Olivier Schwartz  http://orcid.org/0000-0002-0729-1475
Roger Le Grand  http://orcid.org/0000-0002-4928-4484
Rodolphe Thiébaut  http://orcid.org/0000-0002-5235-3962

### Ethics
Cynomolgus macaques (Macaca fascicularis), aged 37–66 months (18 females and 13 males) and originating from Mauritian AAALAC certified breeding centers were used in this study. All animals were housed in IDMIT facilities (CEA, Fontenay-aux-roses), under BSL2 and BSL-3 containment when necessary (Animal facility authorization #D92-032-02, Préfecture des Hauts de Seine, France) and in compliance with European Directive 2010/63/EU, the French regulations and the Standards for Human Care and Use of Laboratory Animals, of the Office for Laboratory Animal Welfare (OLAW, assurance number #A5826-01, US). The protocols were approved by the institutional ethical committee "Comité d'Ethique en Expérimentation Animale du Commissariat à l'Energie Atomique et aux Energies Alternatives" (CEtEA #44) under statement number A20-011. The study was authorized by the "Research, Innovation and Education Ministry" under registration number APAFIS#24434–2020030216532863 v1.

### Decision letter and Author response
Decision letter https://doi.org/10.7554/eLife.75427.sa1
Author response https://doi.org/10.7554/eLife.75427.sa2

## Additional files

### Supplementary files
• Supplementary file 1. Criteria defining neutralization as mechanistic correlate of protection of the effect of the vaccine on new cell infection.

• Supplementary file 2. Model parameters estimated by the model adjusted for groups of intervention.

• Supplementary file 3. Model parameters estimated by profile likelihood.

• Supplementary file 4. Model parameters estimated by the model adjusted for receptor-binding domain (RBD)/ACE2-binding inhibition on beta and for groups on delta.

• Transparent reporting form

### Data availability
No unique reagents were generated for this study. Data that support the findings of this study are provided in the source data files of this paper and gather data from (1) the study [Marlin, Nature Com 2021] used in this analysis, which are also directly available online in the section Source data of this related paper (https://www.nature.com/articles/s41467-021-25382-0#Sec17); (2) the study [Brouwer, Cell 2021] used in this analysis, which are also available from the corresponding authors of the related paper and (3) the study [Corbett, NEJM 2020] used in this analysis, which are also available online in the section Supplementary Material of the related paper, excel file labelled ("Supplementary Appendix 2"). Data from the main study [Marlin, Nature Com 2021] can also be found in the open-access repository Dryad using the following DOI: https://doi.org/10.5061/dryad.1zcrjdfv7. The original code (mlxtran models and R) as well as model definition files including the full list of parameters used are available

and free-of-cost on github (Inria SISTM Team) at the following link: https://github.com/sistm/SARSCoV-2modelingNHP, (copy archived at swh:1:rev:a704c80daebc949434694d3f4441e48293c461cc).

The following dataset was generated:

| Author(s) | Year | Dataset title | Dataset URL | Database and Identifier |
|---|---|---|---|---|
| Alexandre M, Marlin R, Prague M, Coleon S, Kahlaoui N, Cardinaud S, Naninck T, Delache B, Surenaud M, Galhaut M, Dereuddre-Bosquet N, Cavarelli M, Maisonnasse P, Centlivre M, Lacabaratz C, Wiedemann A, Zurawski S, Zurawski G, Schwartz O, Sanders RW, Le Grand R, Levy Y, Thiébaut R | 2022 | Viral loads and antibody, cytokine and T-cell responses in NHPs following vaccination targeting SARS-CoV-2 RBD domain to cells expressing CD40 | https://doi.org/10.5061/dryad.1zcrjdfv7 | Dryad Digital Repository, 10.5061/dryad.1zcrjdfv7 |

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

## Appendix 1

## Model building

In the model presented in the manuscript, we considered the two compartments of the URT, trachea, and nasopharynx, as two distinct compartments (i.e., without transfer of virus between them), as described by *Equation AE1*. In each of them, the viral dynamics are described by a target-cell limited model augmented with a compartment describing the dynamics of the inoculated virus ($V_s$). Moreover, in the statistical model describing the model parameters, the three parameters $\beta$, $\delta$, and $P$ were assumed as jointly estimated between the two compartments, with shared random effects and covariates and considering that parameters $\beta$ and $\delta$ are equal in both trachea and nasopharynx ($\beta^T = \beta^N$, $\delta^T = \delta^N$).

$$\begin{cases} \frac{dT^N}{dt} = -\beta^N V_i^N T^N - \mu\beta^N V_s^N T^N \\ \frac{dI_1^N}{dt} = \beta^N V_i^N T^N + \mu\beta^N V_s^N T^N - kI_1^N \\ \frac{dI_2^N}{dt} = kI_1^N - \delta^N I_2^N \\ \frac{dV_i^N}{dt} = \mu P^N I_2^N - cV_i^N - \beta^N V_i^N T^N \\ \frac{dV_{ni}^N}{dt} = (1-\mu) P^N I_2^N - cV_{ni}^N \\ \frac{dV_s^N}{dt} = -c_i V_s^N - \mu\beta^N V_s^N T^N \end{cases} \qquad \begin{cases} \frac{dT^T}{dt} = -\beta^T V_i^T T^T - \mu\beta^T V_s^T T^T \\ \frac{dI_1^T}{dt} = \beta^T V_i^T T^T + \mu\beta^T V_s^T T^T - kI_1^T \\ \frac{dI_2^T}{dt} = kI_1^T - \delta^T I_2^T \\ \frac{dV_i^T}{dt} = \mu P^T I_2^T - cV_i^T - \beta^T V_i^T T^T \\ \frac{dV_{ni}^T}{dt} = (1-\mu) P^T I_2^T - cV_{ni}^T \\ \frac{dV_s^T}{dt} = -c_i V_s^T - \mu\beta^T V_s^T T^T \end{cases} \qquad \text{(AE1)}$$

Initially, random effects were added on the three parameters. However, taken into consideration identifiability issues that are usually encountered between the viral infectivity ($\beta$) and the viral production ($P$), we decided to remove the possibility of interindividual variability on the parameter $P$. This choice was also driven by multiple model estimations showing less robust estimations when variability was allowed in both parameters $\beta$ and $P$. In particular, the estimate of the viral production was impacted by a ratio between the parameter and its standard error (RSE) higher than 100%.

## Comparison of the parameters between the tracheal and the nasopharyngeal compartments

To decide which of these three parameters were assumed to be equal between the two compartments, all possibilities were tested and compared, using the BICc as selection criteria. As shown in *Appendix 1—table 1*, we started with the model in which all parameters were equal between the two compartments and we progressively relaxed this hypothesis. During this step, no exchange of virions between the two compartments of the URT was possible ($g$=0). Once all models estimated, we kept the one with the lowest value of BICc, meaning with the highest negative difference of BICc compared to the initial model. We identified the model with only the viral production varying between the two compartments as the best one to fit the data.

**Appendix 1—table 1.** Comparison of models evaluating the difference of viral infectivity ($\beta$), loss of infected cells ($\delta$), and viral production ($P$) between the nasopharynx and the trachea.

| Model tested | Statistical model | $\Delta$BICc |
|---|---|---|
| Initial model | $\beta^T = \beta^N$<br>$\delta^N = \delta^T$<br>$P^N = P^T$<br>Variability on $\beta$ and $\delta$ | |
| Model with different $\beta$ | $\boldsymbol{\beta^T \neq \beta^N}$<br>$\delta^N = \delta^T$<br>$P^N = P^T$<br>Variability on $\beta$ and $\delta$ | −17.31 |
| Model with different $\delta$ | $\beta^T = \beta^N$<br>$\boldsymbol{\delta^N \neq \delta^T}$<br>$P^N = P^T$<br>Variability on $\beta$ and $\delta$ | −14.38 |

*Appendix 1—table 1 Continued on next page*

*Appendix 1—table 1 Continued*

| Model tested | Statistical model | ΔBICc |
| --- | --- | --- |
| Model with different $P$ | $\beta^T = \beta^N$<br>$\delta^N = \delta^T$<br>$P^N \neq P^T$<br>Variability on $\beta$ and $\delta$ | **−25.24** |
| Model with different $\beta$ and $\delta$ | $\beta^T \neq \beta^N$<br>$\delta^N \neq \delta^T$<br>$P^N = P^T$<br>Variability on $\beta$ and $\delta$ | −13.00 |
| Model with different $\beta$ and $P$ | $\beta^T \neq \beta^N$<br>$\delta^N = \delta^T$<br>$P^N \neq P^T$<br>Variability on $\beta$ and $\delta$ | −19.19 |
| Model with different $\delta$ and $P$ | $\beta^T = \beta^N$<br>$\delta^N \neq \delta^T$<br>$P^N \neq P^T$<br>Variability on $\beta$ and $\delta$ | −19.47 |
| Model with different $\beta$, $\delta$, and $P$ | $\beta^T \neq \beta^N$<br>$\delta^N \neq \delta^T$<br>$P^N \neq P^T$<br>Variability on $\beta$ and $\delta$ | −13.39 |

## Identification of group effects

Once the structure of the statistical model defined, we tried to identify on which parameters an effect of the group of treatment could be identified and by extension on which biological mechanisms. In this step, we were interested in four parameters: $\beta$, $\delta$, $P$, and $c$, the latter being the clearance of de novo- produced virions. In the study, three groups of treatments were considered as constant categorical covariates: naïve, convalescent, and convalescent vaccinated. We performed a forward selection approach using the BICc as selection criteria to find the best model, using the model without covariate as initial model. At each step the model decreasing the most the value of the BICc is selected and the procedure stops once the BICc does not decrease anymore. At each step of the procedure, the statistical significance of covariate added into the model was verified via a Wald test. As shown in *Appendix 1—table 2*, the selected model identified a group effect on the viral infectivity and the loss rate of infected cells.

**Appendix 1—table 2.** Comparison of models evaluating the adjustment of the viral infectivity ($\beta$), the loss rate of infected cells ($\delta$), the viral production ($P$), and the viral clearance ($c$) for the groups of treatment.
The group of naïve animals is assumed as the group of reference.

| Step | Model tested | Statistical model | $\Delta BICc$ |
|---|---|---|---|
| | Initial model: Model without group effects | $\beta = 10^{\beta_0}$ <br> $\delta = \delta_0$ <br> $P = P_0$ <br> $c = c_0$ | |
| | Model with group effect on $\beta$ | $\beta = 10^{\left(\beta_0 + \phi_{conv}^{\beta} + \phi_{CD40}^{\beta}\right)}$ <br> $\delta = \delta_0$ <br> $P = P_0$ <br> $c = c_0$ | −21.5 |
| | Model with group effect on $\delta$ | $\beta = 10^{\beta_0}$ <br> $\delta = \delta_0 \exp\left(\phi_{conv}^{\delta} + \phi_{CD40}^{\delta}\right)$ <br> $P = P_0$ <br> $c = c_0$ | −16.62 |
| | Model with group effect on $P$ | $\beta = 10^{\beta_0}$ <br> $\delta = \delta_0$ <br> $P = P_0 \exp\left(\phi_{conv}^{P} + \phi_{CD40}^{P}\right)$ <br> $c = c_0$ | +9.68 |
| 1 | Model with group effect on c | $\beta = 10^{\beta_0}$ <br> $\delta = \delta_0$ <br> $P = P_0$ <br> $c = c_0 \exp\left(\phi_{conv}^{c} + \phi_{CD40}^{c}\right)$ | +9.20 |
| | Initial model: Model with group effect on $\beta$ | $\beta = 10^{\left(\beta_0 + \phi_{conv}^{\beta} + \phi_{CD40}^{\beta}\right)}$ <br> $\delta = \delta_0$ <br> $P = P_0$ <br> $c = c_0$ | |
| | Model with group effect on $\beta$ and $\delta$ | $\boldsymbol{\beta = 10^{\left(\beta_0 + \phi_{conv}^{\beta} + \phi_{CD40}^{\beta}\right)}}$ <br> $\boldsymbol{\delta = \delta_0 \exp\left(\phi_{conv}^{\delta} + \phi_{CD40}^{\delta}\right)}$ <br> $P = P_0$ <br> $c = c_0$ | −2.48 |
| | Model with group effect on $\beta$ and $P$ | $\beta = 10^{\left(\beta_0 + \phi_{conv}^{\beta} + \phi_{CD40}^{\beta}\right)}$ <br> $\delta = \delta_0$ <br> $P = P_0 \exp\left(\phi_{conv}^{P} + \phi_{CD40}^{P}\right)$ <br> $c = c_0$ | +12.25 |
| 2 | Model with group effect on $\beta$ and $c$ | $\beta = 10^{\left(\beta_0 + \phi_{conv}^{\beta} + \phi_{CD40}^{\beta}\right)}$ <br> $\delta = \delta_0$ <br> $P = P_0$ <br> $c = c_0 \exp\left(\phi_{conv}^{c} + \phi_{CD40}^{c}\right)$ | +11.97 |
| | Initial model: Model with group effect on $\beta$ and $\delta$ | $\beta = 10^{\left(\beta_0 + \phi_{conv}^{\beta} + \phi_{CD40}^{\beta}\right)}$ <br> $\delta = \delta_0 \exp\left(\phi_{conv}^{\delta} + \phi_{CD40}^{\delta}\right)$ <br> $P = P_0$ <br> $c = c_0$ | |
| | Model with group effect on $\beta$, $\delta$, and $P$ | $\beta = 10^{\left(\beta_0 + \phi_{conv}^{\beta} + \phi_{CD40}^{\beta}\right)}$ <br> $\delta = \delta_0 \exp\left(\phi_{conv}^{\delta} + \phi_{CD40}^{\delta}\right)$ <br> $P = P_0 \exp\left(\phi_{conv}^{P} + \phi_{CD40}^{P}\right)$ <br> $c = c_0$ | +10.88 |
| 3 | Model with group effect on $\beta$, $\delta$, and $c$ | $\beta = 10^{\left(\beta_0 + \phi_{conv}^{\beta} + \phi_{CD40}^{\beta}\right)}$ <br> $\delta = \delta_0 \exp\left(\phi_{conv}^{\delta} + \phi_{CD40}^{\delta}\right)$ <br> $P = P_0$ <br> $c = c_0 \exp\left(\phi_{conv}^{c} + \phi_{CD40}^{c}\right)$ | +11.61 |

Based on all these results, the optimal statistical model with adjustment for groups of treatment was defined as follows:

$$\begin{cases} \log_{10}(\beta_i) = \beta_0 + \phi^{\beta}_{conv} \times I_{i \in conv} + \phi^{\beta}_{CD40} \times I_{i \in CD40} + u^{\beta}_i \\ \log(\delta_i) = \log(\delta_0) + \phi^{\delta}_{conv} \times I_{i \in conv} + \phi^{\delta}_{CD40} \times I_{i \in CD40} + u^{\delta}_i \\ \log(P^N_i) = \log(P_0) \\ P^T_i = P^N_i \times \exp(f^T_P) \end{cases}$$

## Exchange of viruses between the nasopharyngeal and tracheal compartments

Afterward, we tested the possibility of an exchange of free plasma virus from between the two compartments of the URT. We made the hypothesis of a constant first-order exchange and we tested the addition a transfer of virions from nasopharyngeal to tracheal compartments and vice versa, with a migration rate $g_{NT}$ and $g_{TN}$, respectively. To this end, equations of infectious ($V_i$) and non-infectious ($V_{ni}$) viruses in *Equation AE1* between the two compartments were linked as follows:

$$\frac{dV^T_i}{dt} \mapsto \frac{dV^T_i}{dt} - g_{TN}V^T_i + g_{NT}V^N_i \qquad \frac{dV^T_{ni}}{dt} \mapsto \frac{dV^T_{ni}}{dt} - g_{TN}V^T_{ni} + g_{NT}V^N_{ni}$$
$$\frac{dV^N_i}{dt} \mapsto \frac{dV^N_i}{dt} + g_{TN}V^T_i - g_{NT}V^N_i \qquad \frac{dV^N_{ni}}{dt} \mapsto \frac{dV^N_{ni}}{dt} + g_{TN}V^T_{ni} - g_{NT}V^N_{ni} \tag{AE2}$$

with the arrow symbolizing the modification of the equations defined in *Equation AE1* and $g_{NT}$ and $g_{TN}$ being two positive rates. As a first step, we tried to estimate either bidirectional or one of the two unidirectional transfers using the data from the 18 NHPs of the first study described in the main paper. However, data were too spare to bring enough information to get estimations. Consequently, as a second step, additional data were used: two naïve macaques were exposed to the same dose ($1 \times 10^6$ pfu) of SARS-CoV-2 than the 18 NHPs of the main study. However, instead of being inoculated via intra-tracheal (4.5 mL) and intra-nasal (0.5 mL) routes, these latter received inoculum via intra-gastric (4.5 mL) and intra-nasal (0.5 mL) routes. Similar to the main study, the viral gRNA dynamics in both tracheal and nasopharyngeal compartments were repeatedly measured during the 20 days following the challenge (*Figure 2—figure supplement 2E*).

These two additional macaques having not received intra-tracheal inoculum, viral dynamics measured in this same compartment was expected to come from (at least partially) an exchange with the nasopharynx and thus bring information about it. However, having only two macaques without virions inoculated via intra-tracheal route, no enough information were available to totally estimate the model with exchanges. Consequently, these two additional NHPs having similar characteristics than the 18 NHPs involved in the main study, we made the assumption that the viral dynamics in nasopharynx after inoculation and the viral dynamics in the trachea, once the transfer initiated, should be described by the same model (without inoculum in trachea) and those by the same parameters. We expected that the difference of dynamics in trachea between these two set of macaques could allow an estimation of the parameters $g_{TN}$ and/or $g_{NT}$. For that reason, we estimated the model in *Equation AE1* using data from the 18 NHPs of the main study. Then using the data from the two additional NHPs, and assuming all parameters of the model resulting from *Equation AE2* as fixed (see *Appendix 1—table 2*), except $g_{TN}$ and $g_{NT}$, we tried to quantify the transfers of virions.

The estimation of multiple models on those two animals tended to conclude that only a unidirectional transfer of viruses from the nasopharyngeal to the tracheal compartment should be explored, with an estimation of $g_{NT}$ ranging from 0.9 to 2.5 day$^{-1}$. Once these values quantified, we tried to update/re-estimate the model, initially estimated on the 18 NHPs, using only a unidirectional transfer from nasopharynx to trachea and fixing the value of the migration rate at the different values aforementioned. However, all tested values of $g_{NT}$ led irremediably to a degradation of the model with an increase of at least two points of BICc.

An estimation of the parameter $g_{NT}$ by profile likelihood (results not shown) led to a strictly increasing profile of the likelihood (the lower the better) and was thus no more conclusive. Consequently, no exchange of virions were assumed in the final model and the parameters $g_{NT}$ and $g_{TN}$ were fixed at 0 day$^{-1}$.

## Appendix 2

### BICc as selection criteria and multiple testing adjustment

In the case of classic covariate selection approaches using p-values as selection criteria, particular attention must be paid to take into account the dependence of the results on the number tests performed.

Over the years, multiple corrections have been proposed to adjust results for test multiplicity (e.g., Bonferroni correction, Benjamini and Hochberg correction among others).

Although we verified the significance of the covariate selected in our model, our covariate selection approach relies on the BICc. To ensure the robustness of the BICc as selection criterion despite the multiplicity of the tests, we performed an additional simulation work.

We simulated $M$=25 longitudinal variables for 18 individuals and with similar time points than those found on our data, meaning at days 0, 4, 9, and 20 post-infection. Variables were simulated as white-noise random variables such that for the $i$th subject at the $j$th time point, the $m$th variable was defined as $X_{ij}^m \sim \mathfrak{N}\left(0, \sigma^2\right)$, with $m$=1, …, $M$. In our simulations, we tested five values for the variance $\sigma^2$ ranging from 1% to 10% (five variables simulated for each value of $\sigma$).

Assuming these variables as our time-varying covariates, we applied the forward selection approach used in our method by testing each of them in a univariate manner of both $\beta$ and $\delta$.

As shown in *Appendix 2—figure 1*, the 50 models built to evaluate the adjustment of either $\beta$ or $\delta$ for the simulated variables provide similar results in terms of BICc, and thus whatever the value of the standard deviation $\sigma$ used. Consequently, these results appear as quite robust to the multiplicity of the test. Moreover, as expected, adjustments for white-noise random variables depict the degradation of the model in comparison to the model without covariates.

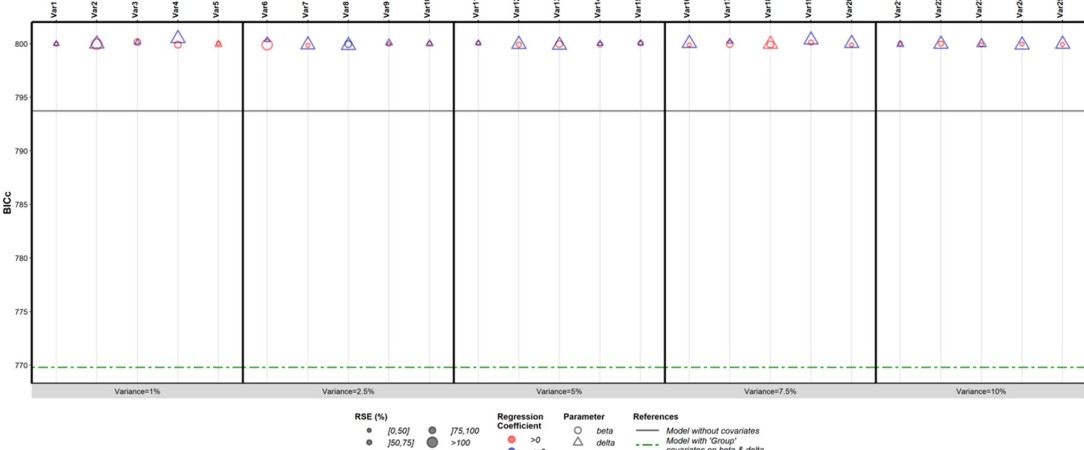

**Appendix 2—figure 1.** Results of the forward selection approach applied on the 25 simulated white-noise random variables. The discrete x-axis represents the different variables and the y-axis represents the values of the corrected Bayesian information criteria (BICc). Circles and triangles correspond to the results obtained with the parameters $\beta$ or $\delta$ adjusted for the variables. The horizontal solid black line represents the value of the BICc obtained with the model without covariates while the horizontal dashed green line highlights the value of the criterion obtained with both $\beta$ and $\delta$ adjusted for the groups of treatment.

### Evaluation of the robustness of the estimation

To evaluate the robustness of the parameter estimates obtained on our models, despite the small number of independent observations, we performed a bootstrap procedure with replacement (*Thai et al., 2014*), for $B$=50 iterations. The bootstrap parameter estimate was calculated as the median of the parameter estimates from the $B$ bootstrap samples while the standard error of each parameter was calculated according to the definition of *Thai et al., 2014*, which means with the SE of the $l$th component of the vector of parameters given by:

$$\hat{SE}_B^l = \sqrt{\frac{1}{B-1}\sum_{b=1}^{B}\left(\hat{\theta}_b^{*(l)} - \hat{\theta}_B^{(l)}\right)^2}$$

with $\theta_b^{*(l)}$ being its estimate obtained at the $b$th iteration of the bootstrap and $\theta_B^{(l)}$ the bootstrap parameter estimate. For each bootstrap sample, we paid attention to keep the 1:1:1 ratio between the three groups of treatment, with six animals selected within each group. Results are reported in *Appendix 2—table 1* and *Appendix 2—table 2*.

**Appendix 2—table 1.** Model parameters for viral dynamics in both the nasopharynx and the trachea estimated by the model adjusted for groups of intervention.
For the bootstrap procedure, 50 iterations were performed.

| Parameter | Meaning | Value [95% CI] | Unit |
|---|---|---|---|
| $\beta$ | Viral infectivity in the naive group ($\times 10^{-6}$) | 0.91 [0.12; 7.03] | (copies/mL)$^{-1}$ day$^{-1}$ |
| | Fold change in the convalescent group | 0.15 [0.04; 0.58] | |
| | Fold change in the Conv-CD40 group | 0.006 [0.001; 0.04] | |
| $\delta$ | Loss rate of infected cells in the naive group | 1.09 [0.74; 1.60] | day$^{-1}$ |
| | Fold change in the convalescent group | 1.70 [1.08; 2.66] | |
| | Fold change in the Conv-CD40 group | 2.00 [0.94; 4.27] | |
| $P^N$ | Viral production rate in the naso. ($\times 10^3$) | 10.1 [1.16; 87.7] | virions (cell day)$^{-1}$ |
| $P^T$ | Viral production rate in the trachea ($\times 10^3$) | 0.86 [0.08; 9.19] | virions (cell day)$^{-1}$ |
| $\alpha_{vlsg}$ | Infected cells and sgRNA viral load ratio | 1.42 [0.99; 2.02] | virions cell$^{-1}$ |
| $k$ | Eclipse rate | 3 | day$^{-1}$ |
| $c$ | Clearance of de novo produced viruses | 3 | day$^{-1}$ |
| $c_I$ | Clearance of inoculum | 20 | day$^{-1}$ |
| $\mu$ | Percentage of infectious viruses | $10^{-3}$ | |
| $T_0^{X,nbc}$ | Initial number of target cells | $1.25 \times 10^5$ (naso.) $2.25 \times 10^4$ (trachea) | cells |
| $Inoc_0$ | Number of virions inoculated | $2.19 \times 10^{10}$ | virions |
| $\omega_\beta$ | SD of random effect on $\log_{10}\beta$ | 0.319 [0.111; 0.527] | |
| $\omega_\delta$ | SD of random effect on $\delta$ | 0.122 [-0.039; 0.283] | |
| $\sigma_{VLn}$ | SD of error model gRNA in naso. | 1.24 [0.96; 1.51] | |
| $\sigma_{VLt}$ | SD of error model gRNA in trachea | 1.09 [0.92; 1.26] | |
| $\sigma_{sgVLn}$ | SD of error model sgRNA in naso | 1.35 [1.08; 1.61] | |
| $\sigma_{sgVLt}$ | SD of error model sgRNA in trachea | 1.53 [1.15; 1.92] | |

**Appendix 2—table 2.** Model parameters for viral dynamics in both the nasopharynx and the trachea estimated by the model with the viral infectivity adjusted for ACE2-RBD-binding inhibition and the loss rate of infected cells adjusted for the group of treatment.
For the bootstrap procedure, 50 iterations were performed.

| Parameters | Meaning | Value [95% CI] | Unit |
|---|---|---|---|
| $\beta$ | Infection rate with ECLRBD = 0 AU ($\times 10^{-8}$) | 0.82 [0.13; 5.13] | (copies/mL)$^{-1}$ day$^{-1}$ |
| | Fold $\Delta\mathbf{ECLRBD} = 10^3$ AU | 1.017 [1.012; 1.022] | |

*Appendix 2—table 2 Continued on next page*

*Appendix 2—table 2 Continued*

| Parameters | Meaning | Value [95% CI] | Unit |
|---|---|---|---|
| | Loss rate of infected cells | 1.02 [0.80; 1.30] | day$^{-1}$ |
| $\delta$ | Fold change in the convalescent group | 1.74 [1.24; 2.46] | |
| | Fold change in the Conv-CD40 group | 2.17 [0.82; 5.74] | |
| $P^N$ | Viral production rate in the naso. ($\times 10^3$) | 8.92 [0.42; 191] | virions (cell day)$^{-1}$ |
| $P^T$ | Viral production rate in the trachea ($\times 10^3$) | 0.62 [0.02; 19.7] | virions (cell day)$^{-1}$ |
| $\alpha_{vlsg}$ | Infected cells and sgRNA viral load ratio | 1.32 [0.91; 1.90] | virions cell$^{-1}$ |
| $k$ | Eclipse rate | 3 | day$^{-1}$ |
| $c$ | Clearance of de novo produced viruses | 3 | day$^{-1}$ |
| $c_I$ | Clearance of inoculum | 20 | day$^{-1}$ |
| $\mu$ | Percentage of infectious viruses | $10^{-3}$ | |
| $T_0^{X,nbc}$ | Initial number of target cells | $1.25 \times 10^5$ (naso.) $2.25 \times 10^4$ (trachea) | cells |
| $Inoc_0$ | Number of virions inoculated | $2.19 \times 10^{10}$ | virions |
| $\omega_\beta$ | SD of random effect on $\log_{10} \beta$ | 0.205 [0.011; 0.399] | |
| $\omega_\delta$ | SD of random effect on $\delta$ | 0.079 [-0.092; 0.250] | |
| $\sigma_{VLn}$ | SD of error model gRNA in naso. | 1.13 [0.90; 1.36] | |
| $\sigma_{VLt}$ | SD of error model gRNA in trachea | 1.27 [1.07; 1.48] | |
| $\sigma_{sgVLn}$ | SD of error model sgRNA in naso | 1.62 [1.30; 1.94] | |
| $\sigma_{sgVLt}$ | SD of error model sgRNA in trachea | 1.36 [1.15; 1.56] | |

