## [Editor Report]

This work should be of interest to a broad readership in infectious diseases, especially those people interested in modeling of infections. It combines statistical and mechanistic modeling to find assayable correlates of immunity for vaccines. This method could be relevant to many diseases or vaccines, although the particular markers identified here likely will be limited in their generalizability.

---

## [Decision Letter]

**Decision letter after peer review:**

Thank you for submitting your article "SARS-CoV-2 mechanistic correlates of protection: insight from modelling response to vaccines" for consideration by *eLife*. Your article has been reviewed by 3 peer reviewers, and the evaluation has been overseen by a Reviewing Editor and Miles Davenport as the Senior Editor. The reviewers have opted to remain anonymous.

The reviewers have discussed their reviews with one another, and the Reviewing Editor has drafted this to help you prepare a revised submission. The reviewers appreciated the developed approach and analyses, but had several concerns regarding the robustness of the methods and estimates, as well as the conclusions drawn from it. In particular, they identified the following main points that would need to be addressed in order to advance the study:

Essential revisions:

(1) A more detailed explanation of the used methods. This especially applies to the selection of biomarkers, as well as addressing the potential risk of overfitting in case of the mechanistic model.

(2) A revision of the title to specify that the analysis is limited to non-human primates in the light of the specific comments by the reviewers below.

(3) A more detailed discussion of the limitations of the study and reconsideration of the conclusions with regard to the interpretability of the non-human data for the human situation and possible caveats in the analysis. An inclusion of human data sets as suggested by reviewer 2 is not necessarily warranted. But a more detailed/extended discussion of the literature on correlates of protection should be included. Please also have a look at the detailed suggestions and links provided by reviewer 2.

We would welcome a substantially revised version of the manuscript addressing all of the issues mentioned by the reviewers. Please also see the detailed suggestions that are made by the reviewers below, and we hope you will find them helpful in this regard.

*Reviewer #1 (Recommendations for the authors):*

The model used seems adequate, but there are several modeling options that could be better justified. This would help the reader understand the modeling approach better. In some cases, this is simply a case of explaining earlier why you are making the choice that you make. For example, why do you model infection and non-infectious virus (Line 119)? I think this is to fit both virus and infected cells, but this is not clear when you present this choice, which then makes it a little unclear in line 124 when you say that sgRNA is proportional to infected cells. Again, why this choice? Wouldn't it also be reasonable to consider gRNA plus sgRNA to be proportional to infected cells? Another modeling choice that could be better explained is why you consider the inoculum virus separately (its own compartment and its own clearance rate)? In Line 401 you say, "to be able to distinguish", but why is that needed?

When you describe the model in the Results section, it would also be important to say something about the coupling of URT and trachea – this is only at the end of the model description in the Methods, but since you mention the two compartments in the results, the reader (who may not read the Methods first) may be confused.

Some more questions about the model. Is the clearance rate the same for infectious and non-infections virus? Line 470, you use profile likelihood for these parameters, what about the values of the other parameters when you are calculating these profiles? Line 511, this seems a little bit circular: you estimate all parameters with gs zero, then fix all parameters and try to estimate the two gs. But perhaps the other parameters would be different with g not zero?

The process for automatic selection of biomarkers also needs a little more clarification. It seems to be a generalization of stepwise analyses, but the issue of the combination of two lower ranked markers potentially being better than a higher ranked one is not mentioned or discussed. This is an issue because each parameter is tried individually. Also, you say (line 526) that it is possible to add time-varying covariates in this methodology, but you don't say anything about how that is done (it is not trivial), and in fact you almost don't say anything about results with time-varying covariates. Some of these aspects may also be relevant for discussion or limitation of the approach.

Line 238, even though you say that the mechanistic model was better, that is not what is presented in Table S1, where the group effect model is better. Please clarify. Also, in line 246 you state "no additional effect", but Table S1 presents results for δ. Also, Table S1 is not mentioned in this section of the manuscript.

I may have missed it, but it seems that in the "Results" nothing is said about time varying markers…

Something seems strange in some graphs in figure 2b and S8. In some cases, the mean value appears to be below every individual value; or sometimes above every individual value (for example, in the naïve). Also, are the thin lines individual predictions as stated, why do these lines have kinks instead of being smooth and do they really go through every data point in every animal (this is clear for example in the naïve, where the thin lines are clearly visible, are not smooth and go through every data point)?

*Reviewer #2 (Recommendations for the authors):*

– What the authors state the main contribution as (new framework of modeling) and what I found to be the main conclusion (vaccine-induced antibody binding indicates protectiveness) are different. Because the model and analytical approach is not particularly novel, I would suggest redesigning the manuscript to highlight the data and biological conclusions.

– Getting into the business of claiming there is some magical "correlate" is very tricky. I don't feel that this is what we need to understand the infection or response to vaccination, and the work in this manuscript does not assess sufficient data to make broad claims about any clinically-meaningful correlate. With as much data as there is in the literature, a validation using human data is warranted.

– In the introduction, the authors reference studies that suggest "binding antibodies to SARS-CoV-2 and in vitro neutralization of virus infection are clearly associated with protection". This seems to be the same conclusion that the authors came to and makes me question: what is new here?

– A more robust introduction that highlights the current conclusions and limitations of models in the literature is needed, particularly because the model presented is only a minor modification of those models. In addition, the model lacks immune dynamics, which would make its usefulness limited.

– In addition to the point above, while the model is a standard, previously published viral dynamics model with minor modifications, it was not adequately described for the broad readership of *eLife*. Perhaps I missed it, but I couldn't find where the terms were statistically justified?

– Several models have shown that viral load data, even when defined between compartments, is insufficient to distinguish transport between the nasal passages and trachea (e.g., Khan et al. Viruses, Ke et al. MedRxiv, Pinky et al. PCB, among others). In light of these studies, this part of the model seems unnecessary and one unsupported by the data. In addition, these other studies have also shown that viral and immune parameters are distinct in the nasopharynx and trachea. In the authors' work, the data were joined and no effective comparison was made, which may cloud the parameter estimates and conclusions. Minimally, a discussion and reference of these published works is needed.

– Another limitation of the model and the data is that viral loads in the nasopharynx can be similar when disease is not. Can the approach here be used to assess vaccine efficacy (in terms of reducing disease)?

– Curiously, CD8 T cells were measured in the serum, but this type of measurement tends to not reflect the events in the tissue where resident T cells are thought to be important. There were some comments made about the infected cell clearance rate, but it did not seem that the data were used to evaluate the parameter estimates or model conclusions.

– Macaques, unfortunately, are not a great model to assess the individual heterogeneity observed in humans. Dissecting the heterogeneity from different sources would be greatly beneficial, but it seems naïve to assume that this can be done using animal models. I suggest using a human dataset, where there are many to choose from in the literature, or artificially creating one from human data and see if the model can still perform.

Possible data sets and information:

(i) CDC list of studies of correlates of protection, reinfection studies, etc.:

https://www.cdc.gov/coronavirus/2019-ncov/science/science-briefs/vaccine-induced-immunity.html

(ii) These quantify antibody binding + other immunology: https://www.science.org/doi/full/10.1126/science.abm3425

https://www.nature.com/articles/s41586-021-03738-2?r=artikellink

https://www.sciencedirect.com/science/article/pii/S0092867421007066

(iii) One that has nice temporal data of lots of different antibodies and cells: https://www.science.org/doi/10.1126/science.abf4063

(iv) One that shows antibody levels + whether the macaque was protected: https://www.nature.com/articles/s41586-020-03041-6

– In reference to Table S1, it is stated that vaccinated and unvaccinated animals could be distinguished. Why is this the goal? We would know who is/isn't vaccinated – a better question is who among the vaccinated would not be protected from infection and/or disease.

– In Line 189-191, the authors state "both specific antibodies and specific CD8^+^ T cells are mechanisms commonly considered important for killing infected cells. We retained the anti-RBD binding IgG Ab that were positively associated to the increase of the loss of infected cells.". Unfortunately, this statement is incorrect as antibodies cannot kill infected cells. Antibodies neutralize virus so that cells are not infected, but they cannot kill cells.

– Prior studies have found that antibodies have a limited role during infection (e.g., see Goyal et al. Viruses). How does this fit into the results?

– Figure 2 is difficult to assess what is data and what is the model, and the model fit is difficult to see. It seems a though the 95% confidence intervals (gray shading) are also not indicative the model. Why are there multiple peaks and does the model capture that?

– The discussion could be improved. It lacked discussion of how the work fits into the literature, particularly other published SARS-CoV-2 models, how the correlate might be used in the clinic, etc. In addition, a discussion about the differences in dynamics within the nasopharynx and trachea in vaccinated individuals would have been interesting.

– It's unclear to me why the reproduction number R was under 1 for the alphaCD40.RBD group when there was replication for several days in this group.

*Reviewer #3 (Recommendations for the authors):*

First, a suggestion regarding my overfitting concern:

• The number of non-human primates in each arm of these studies is understandably limited, but cross-validation could be used to reduce the extent of overfitting even with limited replicates.

Next, two suggestions to enhance clarity:

• The rationale for the use of profile likelihood should be made explicit. Why do a few parameters using this approach and then later do rest with full maximum likelihood? Is it a limitation of the optimization software?

• In the model, why separately track virions from the inoculum versus virions generated in host? If it is just to allow for different clearance rates, why do we expect these to be different?

---

## [Author Response]

Essential revisions:1.) A more detailed explanation of the used methods. This especially applies to the selection of biomarkers, as well as addressing the potential risk of overfitting in case of the mechanistic model.

As pointed out by the reviewers, the first version of the article was somewhat poor in explanations about the methods used. Accordingly, we have described the modeling by adding some more precision. In particular, we have added more explanations about:

1. The choice of the mechanistic model used and more specifically about

a. The distinction of infectious and non-infectious viruses (see Reviewer 1, remarks 1a, 3a)

b. The distinction of the inoculated and the produced de novo virions (see Reviewer 1, remark 1b)

c. About our modeling choice to integrate subgenomic data in the observation model (see Reviewer 1, remark 1a)

d. About the coupling of URT compartments (the trachea and nasopharynx) and the exchange of virions between these two compartments (see Reviewer 1, remarks 2 and 3c and reviewer 2, remark 6).

2. The algorithm of covariate selection that we implemented, and more specifically about

a. The type of algorithm used and its robustness (see Reviewer 1 remark 4a, Reviewer 3 remark 1)

b. The difference of the algorithm compared to classical algorithms to incorporate time-varying variables (see Reviewer 1, remark 4b)

c. The limitation of the proposed algorithm (see Reviewer 1, remark 4b)

2.) A revision of the title to specify that the analysis is limited to non-human primates in the light of the specific comments by the reviewers below.

As requested, we modified the title of the article to specify its current limitation to non-human primates. The proposed new title is “Modelling the response to vaccine in Non-Human Primates to define SARS-CoV-2 mechanistic correlates of protection”.

3.) A more detailed discussion of the limitations of the study and reconsideration of the conclusions with regard to the interpretability of the non-human data for the human situation and possible caveats in the analysis. An inclusion of human data sets as suggested by reviewer 2 is not necessarily warranted. But a more detailed/extended discussion of the literature on correlates of protection should be included. Please also have a look at the detailed suggestions and links provided by reviewer 2.

No application on human data has been included in this revised version of the article. However, a particular attention has been paid to information provided by reviewer 2 to enrich the discussion about the proposed approach, whether about its limitations or its potential extensions.

Reviewer #1 (Recommendations for the authors):The model used seems adequate, but there are several modeling options that could be better justified. This would help the reader understand the modeling approach better. In some cases, this is simply a case of explaining earlier why you are making the choice that you make. For example, why do you model infection and non-infectious virus (Line 119)? I think this is to fit both virus and infected cells, but this is not clear when you present this choice, which then makes it a little unclear in line 124 when you say that sgRNA is proportional to infected cells. Again, why this choice? Wouldn't it also be reasonable to consider gRNA plus sgRNA to be proportional to infected cells?

About infectious and non-infectious virus:

The SARS-CoV-2 virus exists in both infectious and non-infectious form (1,2) with a proportion μ of virus able to infect new target cells relatively low compared to non-infectious one. According to (3), this ratio can be quantified with a lower bound ranging from 10^-5^ and 10^-4^. Expert opinion as well as the evaluation of this same ratio on similar NHP than those used on our main study (unpublished results) led us to the value of 10^-3^. The final choice of the value of the parameter μ was guided by the comparison of the results obtained for μ=10^-3^ and 10^-4^. Indeed, estimations appeared as more stable with the highest value.

Nevertheless, we agree with the fact that outside the ODE system, compartments of virions are always used as the sum of *V_i_* and *V_ni_*. Consequently, the model could be re-written assuming a single compartment for plasma free viruses (*V*), paying attention to replace the terms β*V_i_T* by βμVT to keep the ratio between infectious and non-infectious viral particles. We decided to write the model with distinct compartments between *V_i_* and *V_ni_* for a better visual understanding of the model and of the parameter μ.

We add the justification about the modelling of both infectious and non-infectious viruses in the manuscript (see Results page 7, Lines 121-122 and Methods, page 21, Lines 453-457).

[Addition applied in section Results, page 7, Lines 121-122]

“Although a single compartment for de novo produced viruses (*V*) could be mathematically considered, two distinct ODE compartments were assumed for a better understanding of the model.”

[Addition applied in section Methods, page 21, Lines 453-457]

“… remaining proportion of virions is non-infectious (*V_ni_*). Mathematically, a single compartment (*V*) for de novo produced virions could be considered in the model, with *μV* and *(1- μ)V* representing the respective contributions of infectious and non-infectious viruses to the biological mechanisms. However, to have a better visual understanding of the distinction between the two types of viruses, we wrote the model with distinct compartments, *V_i_* and *V_ni_*.”

About subgenomic RNA:

In our model, we assume the subgenomic RNA (sgRNA) as proportional to infected cells. First, sgRNA is only transcribed in infected cells (4). Second, the use of this additional marker allowed us to improve the identifiability of model parameters. According to the paper of Miao et al. (2011) (5), the use of observations for both virus and infected cell compartments allows to counteract structural identifiability issue in this type of target-cell limited models, in particular between the parameters of infectivity (β) and viral production (*P*). In accordance with this paper, we tested different parametrizations to jointly describe gRNA and sgRNA in the observation model. In particular, we compared the use of the two following observation models:

(a) with gRNA defined by (*V_i_ + V_ni_ + V_s_*) as total viral load and sgRNA, resulting from viral replication, as proportional to (*V_i_ + V_ni_*),

(b) with gRNA defined by (*V_i_ + V_ni_ + V_s_*) and sgRNA as proportional to (*I_1_ + I_2_*).

As expected, the observation model (a) was impacted by identifiability issues compelling us to fix either β or *P*, while the estimation of these two parameters was possible with the model (b). Moreover, this latter led to better values of BIC (-25 points of BICc).

As proposed by the reviewer, another possibility could have been to define the sum of gRNA and sgRNA as proportional to infected cells assuming gRNA characterized by (*V_i_ + V_ni_ + V_s_*) and gRNA + sgRNA by (*I_1_ + I_2_*). But we did not try this solution.

To better explain this choice of modelling, we modified the manuscript as follows (section Results, page 7, Lines 133-135 and section Methods, page 24-25, Lines 508-521).

[Addition applied in section Results, page 7, Lines 133-135]

“We assumed that gRNA and sgRNA were proportional to the free virus and the infected cells, respectively. This modeling choice relied on both biological and mathematical reasons (see the section methods for more details).”

[Addition applied in section Methods, page 24-25, Lines 508-521]

“For the observation model, we jointly described genomic and subgenomic viral loads in the two compartments of the URT. We defined genomic viral load, which characterizes the total viral load observed in a compartment (nasopharynx or trachea), as the sum of inoculated virions (*V_s_*), infectious (*V_i_*) and non-infectious virions (*V_ni_*). The sgRNA was described as proportional to the infected cells (*I_1_ + I_2_*). This choice was driven by two main reasons. First, sgRNA is only transcribed in infected cells (76). Second, as described by Miao et al. (2011) (77), to overcome identifiability issues between the parameters β and *P* typically observed in target-cell limited models. The comparison of the two observation models describing sgRNA as either proportional to virions produced de novo (*V_i_ + V_ni_*) or proportional to infected cells (*I_1_ + I_2_*) confirmed this conclusion. In addition to a better BICc value (-25 points) compared with the first model, the second one allowed the estimation of both β and *P* by counteracting the identifiability problems faced with the first model (results not shown). Accordingly, the log_10_-transformed gRNA and sgRNA of the *i-*th animal at the *j-*th time point in compartment X (nasopharynx or trachea), denoted gRNA_ij_^X^ and sgRNA_ij_^X^ respectively, were jointly described by the following equations …”

Another modeling choice that could be better explained is why you consider the inoculum virus separately (its own compartment and its own clearance rate)? In Line 401 you say, "to be able to distinguish", but why is that needed?

In our model, we distinguished between the inoculated virus and the virus produced de novo by the infected cells by including a compartment *V_s_*. This choice was guided by requests of immunologists for a better understanding of the dynamics of the inoculum. Moreover, this allowed us to use all the information provided by preclinical studies and avoided to estimate or randomly fixe the initial conditions for the V_i_ and V_ni_ compartments, as usually done. Finally, this modeling choice allowed to differentiate the elimination rate of the two virus types. In particular, the inoculum corresponds to a high number of locally inoculated virions, a large fraction of which can be eliminated faster than de novo produced virions due to the experimental conditions (swallowing and natural downstream influx). As described in the paper, using profile likelihood, we showed an improvement in the estimation of model (gain of 5 points of likelihood) when distinct values for the clearance of inoculated and de novo produced virions were considered, leading to a faster elimination of the inoculum, as expected.

To better explain the introduction of the inoculum as ODE compartment, the manuscript was modified as follows (section Methods, page 20, Lines 440-448 and pages 21, Lines 458-465).

[Addition applied in section Methods, page 20, Line 440-448]

“We completed the model by adding a compartment for the inoculum that distinguishes the injected virus (*V_s_*) from the virus produced de novo (*V_i_* and *V_ni_*). To our knowledge, this distinction has not been proposed in any previous work. Two main reasons led us to make this choice. First, it allowed us to study the dynamics of the inoculum, in particular during the early phase of viral RNA load dynamics. Second, as described in more detail below, it gave us the opportunity to use all the information provided by the preclinical studies, such as the known number of inoculated virions, to define the initial conditions of the ODE model rather than estimating or randomly fixing them for *V_i_* and *V_ni_*, as is usually done.”

[Addition applied in section Methods, pages 21, Line 458-465]

“Finally, virions produced de novo and those from the inoculum are cleared at rates ***c*** and ***c_i_*,** respectively. Distinct clearances were considered to account for the effects of experimental conditions on viral dynamics. In particular, it is hypothesized that, animals being locally infected with large numbers of virions, a large proportion of it is assumed to be rapidly eliminated by swallowing and natural downstream influx, in contrast to the de novo produced virions. However, it is important to keep in mind that this distinction was possible because of the controlled experimental conditions performed in animals, (i.e., exact timing and amount of inoculated virus known, and frequent monitoring during the early phase of the viral dynamics).”

When you describe the model in the Results section, it would also be important to say something about the coupling of URT and trachea – this is only at the end of the model description in the Methods, but since you mention the two compartments in the results, the reader (who may not read the Methods first) may be confused.

We have reformulated the presentation of the two compartments (trachea and nasopharynx) in the Results section to better insist on the coupled modelling earlier in the paper. Moreover, we have considered the comment of Reviewer 2 (see comment 6 below) to justify the absence of viral exchange between the two compartments. The manuscript was modified accordingly (section Results, page 6, Lines 107-109 and page 7, Lines 124-133, and section Methods page 28, Lines 591-596).

[Addition/modification applied in section Results, page 6, Lines 107-109]

“The viral dynamics during primary infections were characterized by a peak in genomic RNA (gRNA) production three days post-infection in both tracheal and nasopharyngeal compartments, followed by …”

[Addition/modification applied in section Results, page 7, Lines 124-133]

“In both compartments of the upper respiratory tract (URT), the trachea and nasopharynx, viral dynamics were distinctively described by this model (Figure 2A). Viral exchange between the two compartments was tested (either from the nasopharynx to the trachea or vice versa). However, as described in the literature (28, 42, 43) and demonstrated by the additional modeling work in Appendix 1 “Model building”, viral transport within the respiratory tract plays a negligible role in viral kinetics compared with viral clearance. Consequently, no exchange was considered in the model. Using the gRNA and sgRNA viral loads, we jointly estimated (i.e., shared random effects and covariates) the viral infectivity (β), the viral production (*P*), and the loss rate of infected cells (δ) in the two compartments.”

[Addition applied in section Methods, page 28, Lines 591-596]

“The possibility of migration of free plasma virus between the nasopharynx and the trachea was tested. However, as widely described in the literature, the transport of viral particles within the respiratory tract is negligible in the viral dynamics and is difficult to estimate. The reader can refer to the Appendix 1 “Model building” for an additional modelling work conducted to estimate this exchange and provided the same conclusion. Accordingly, the two compartments of the URT were assumed as distinct in our model.”

Some more questions about the model. Is the clearance rate the same for infectious and non-infections virus?

The clearance rate for infectious and non-infectious viruses is the same, in both URT compartments (c=20 day^-1^). Due to identifiability issues, we were not able to identify distinct clearances, although this could be biologically relevant under certain conditions. Because this point was unclear to the reviewer, we modified the manuscript to clearly mention it in the paper (section Results, page 7, Lines 135-139 and section Methods page 22, Line 468).

[Addition applied in section Results, page 7, Lines 135-139]

“Due to identifiability issues, the duration of the eclipse phase (*1/k*), the clearance of free viruses from the inoculum (*c_i_*) and produced de novo (*c*) were estimated separately by profile likelihood and assumed to be identical in the two compartments of the URT. In addition, infectious and non-infectious viruses were assumed to be cleared at the same rate.”

[Addition/modification applied in section Methods, page 21, Line 466]

“Because of identifiability issues, similar clearances for infectious and non-infectious viruses were used. Accordingly, …”

Line 470, you use profile likelihood for these parameters, what about the values of the other parameters when you are calculating these profiles?

Because of identifiability issues, we performed profile likelihood to determine appropriate values for the three parameters *k*, *c,* and *c_i_.* Profile likelihood consists in sequentially testing different combinations of values of the parameters of interest by fixing them in the model and optimizing their values by choosing the combination that maximizes the likelihood. Accordingly, for each tested combination, all unfixed parameters were estimated here by the SAEM algorithm. Since we are only interested in the maximum likelihood estimates, we did not look at the value of the estimated parameters for each combination, but only for the one that yields the highest log-likelihood (see section Methods, page 26 lines 554-559).

[Addition applied in section Methods, page 26, Lines 554-559]

“The profile likelihood consists in defining a grid of values for the parameters to be evaluated and sequentially fixing these parameters to one of these combinations of values. The model and all the parameters that are not fixed are then estimated by maximizing the log-likelihood. In this process, all parameters that are assumed to be fixed in the model (i.e. *μ* and the initial conditions) are held fixed. Finally, the optimal set of parameters is chosen as the one optimizing the log-likelihood. Although the available data ….”

Line 511, this seems a little bit circular: you estimate all parameters with gs zero, then fix all parameters and try to estimate the two gs. But perhaps the other parameters would be different with g not zero?

We understand the reviewer’s concerns on this point. Nevertheless, the estimation was not completely circular as different data were used to estimate the parameters of the model in the absence of virus exchange and to estimate *g*. While the 18 NHPs from the main study were used in the first case, similar data obtained for two additional naive NHPs that had not received intra-tracheal inoculum were used to quantify exchanges. Accordingly, we assumed that their viral loads observed in the trachea should come from an exchange with the nasopharynx and thus should provide information to estimate *g*. However, because of the small number of new NHPs, we did not have enough observations to fully re-estimate the model with exchange. Consequently, hypothesizing that the two naïve NHPs had similar physiological characteristics to the 18 others, we fixed the model parameters to the value found without exchange and we estimated *g* only*.* In a second time, we tried to estimate the global model directly with the 20 NHPs but results were not more conclusive, with *g* tending to zero.

In response to this comment, to the previous comment 2 and to the comment 6 of the Reviewer 2, we transferred the sub-section “Methods/Exchange of viruses between nasopharynx and trachea compartments” in Appendix 1 “Model building” and we mentioned this file in section Methods (page 28, Lines 594).

The process for automatic selection of biomarkers also needs a little more clarification. It seems to be a generalization of stepwise analyses, but the issue of the combination of two lower ranked markers potentially being better than a higher ranked one is not mentioned or discussed. This is an issue because each parameter is tried individually.

The algorithm implemented for biomarker selection is not exactly a stepwise algorithm as it involves only a forward selection procedure. As mentioned by the reviewer, the case in which two lower ranked markers are better than one higher ranked marker was not mentioned in the manuscript. The use of stepwise selection (i.e. a combination of forward selection and backward elimination) could reduce the number of these cases. Although we could theoretically have implemented stepwise selection, we did not integrate backward elimination in the present work for two practical reasons. The first one is the computational time required to run the algorithm, since a large number of time-varying markers (more than 50) were tested univariately on all parameters of interest. Second, the proposed method for selecting covariates relies not only on a single selection criterion (here, the best fit criterion), as is usually the case, but also on the other two other criteria that define a mCoP (better capture of interindividual variability and replace the group of intervention). In our specific applications, using BIC as selection criterion (rather than p-values) and the selection stopping after two steps, backward elimination would have been useless. Nevertheless, we manually checked the statistical significance of all added covariates at each step. Finally, it is important to note that we are not interested in finding “the” CoP (it is not a biological paper), but we want to find a robust CoP with the good statistical properties.

For sake of simplicity, we merged the modifications related to this comment with those related to the next comment. Both comments relate to the algorithm for selecting covariates.

Also, you say (line 526) that it is possible to add time-varying covariates in this methodology, but you don't say anything about how that is done (it is not trivial), and in fact you almost don't say anything about results with time-varying covariates. Some of these aspects may also be relevant for discussion or limitation of the approach.

We agree that the explanations about time-varying variables and the application of the covariate selection method to them were not entirely clear and sufficiently emphasized in the manuscript. We modified the paper to take these points more into account. In particular, we mentioned:

a) the lack of algorithms for selecting time-varying covariates already implemented in software such as Monolix (see section Methods page 29, lines 602-611).

b) the use of linear interpolation to incorporate the dynamics of the markers as a time-varying covariate in the statistical model (see section Methods, pages 30-31, lines 638-672).

We also added a table in the supplementary files with the parameter estimates of the model in which viral infectivity was adjusted for the time-varying dynamics of antibodies inhibiting the attachment of ACE2 and RBD (see Table 2 of this document, referred as Supplementary file 4 in the manuscript). Finally, we extended the discussion to mention the limitations of the proposed algorithm (see section Discussion, page 16, lines 344-349).

[Addition/modification applied in section Methods, page 29, Lines 602-611]

“… capture this effect. Nowadays, many methods for selecting constant covariates already exist (80) and are implemented in software such as Monolix. However, these latter do not allow time-varying covariates. In this section, we present the algorithm we implemented to select time-varying covariates. We proposed a classical stepwise data-driven automatic covariate modeling method (Figure S10). However, initially implemented to select covariates from more than 50 biomarkers, computational time restricted us to consider only a forward selection procedure. Nevertheless, the method can be easily extended to classical stepwise selection in which both forward selection and backward elimination are performed sequentially. In particular, this extension could contribute to reduce the occurrence of the case in which the combination of two lower ranked markers might be better than a single higher ranked marker. Although the method was developed for time-varying covariates, it can also be applied to constant covariates.”

[Addition applied in section Methods, pages 30-31, Lines 638-672]

“Modelling hypothesis for time-dependent covariates in our application

Using a population-based approach to estimate our mechanistic model and similar to the adjustment of the model for constant covariates (e.g., groups of intervention), time-varying covariates are incorporated into the statistical model as individual-specific explanatory variables in the mixed-effects models. The identification of antibodies inhibiting the attachment of the RBD domain to the ACE2 receptor (ECLRBD) as the first time-varying CoP led to the definition of the time-varying viral infectivity for the *i-*th animal as described in Equation (5), while the selection anti-RBD IgG-binding antibodies (IggRBD) led to the elimination rate of infected cells given in Equation (6).

[Addition/modification applied in section Discussion, pages 16, Lines 344-349]

…that best fit the data. Nevertheless, many modeling choices for the statistical model were made in this approach and more theoretical work evaluating the robustness of the results in their regards may be relevant for future works. In particular, we could relax the constraint of linear interpolation of marker dynamics by using simple regression models, allowing in the same time the integration of error model to account for measurement error for time-varying covariates (63-65). Moreover, by construction, ….”

Line 238, even though you say that the mechanistic model was better, that is not what is presented in Table S1, where the group effect model is better. Please clarify.

We thank the reviewer for this relevant comment. Indeed, some information is missing in the table to fully understand the results. For the first criterion, we have reported the BICc values for the model without covariates (i.e., the initial model in the algorithm), for the model adjusted for group effects on β and δ, except for the third study, and for the model adjusted only for the marker, i.e., without group effect on δ. We added the BICc value of the model with β adjusted for the marker and δ adjusted for the groups of treatment in Table S1 (see below; labelled now Supplementary file 1) to clarify the results. In addition, as described in the algorithm, and in response to comment (4a), the algorithm is based on three criteria, and criterion 1 (best fit) aims at selecting the marker with the optimal value compared with the model without covariates or adjusted for immune markers. The main purpose of the comparison with the model adjusted for groups of treatment is to quantify the gap between the two models. As this is not the case here, the selection of two markers might be necessary to perform better than group effects.

The groups of treatment essentially intervene for the other two criteria: replacement of the effect and explanation of interindividual variability.

Also, in line 246 you state "no additional effect", but Table S1 presents results for δ. Also, Table S1 is not mentioned in this section of the manuscript.

An effect of the group of treatment was well identified only for the parameter β for the third dataset. The information on the parameter δ provided in Table S1 (criterion 3) refers to the interindividual variability of the parameter, random effects being considered on both β and δ. We included this information in the Table (see below) to verify that adjusting the model for β does not severely degrade the estimation of the variance of random effects on δ.

I may have missed it, but it seems that in the "Results" nothing is said about time varying markers…

In the Results section, time-varying markers were indeed not mentioned and we thank the reviewer for this relevant comment. We have corrected the manuscript accordingly (see section Results page 10, lines 195-200, and page 10-11, lines 209-214).

[Addition/modification applied in section Results, pages 10, Lines 195-200]

“To this aim, we performed a systematic screening by adjusting the model for each marker, and we compared these new models with the model without covariates and with the model adjusted for the groups. In particular, our approach allowed us to benefit from all the information provided by the overall dynamics of the immunological markers after the exposure by integrating them as time-varying covariates (see supplemental information for a detailed description of the algorithm) …”

[Addition applied in section Results, pages 10-11, Lines 209-214]

“… Finally, we looked at the estimated viral infectivity according to the binding inhibition assay in each animal. A positive dependence was found between the viral infectivity and the RBD-ACE2 binding inhibition measure, linking an increase of 10^3^ AU of the marker, whether over time or between animals, with an increase of 1.8% (95CI% [1.2%; 2.3%]) of the viral infectivity (see Supplementary file 4). Accordingly, the values at the time of exposure were not overlapping at all, distinguishing clearly the vaccinated and unvaccinated animals (see Figure 4C).”

Something seems strange in some graphs in figure 2b and S8. In some cases, the mean value appears to be below every individual value; or sometimes above every individual value (for example, in the naïve). Also, are the thin lines individual predictions as stated, why do these lines have kinks instead of being smooth and do they really go through every data point in every animal (this is clear for example in the naïve, where the thin lines are clearly visible, are not smooth and go through every data point)?

We thank the reviewer for these helpful comments. Indeed, these figures were very confusing.

In these figures, it is important to note that observations were represented by thick dashed lines (mean value) and shaded areas (interindividual variability), while the predicted data were represented by thin solid lines. Individual predictions were given by thin solid lines. These latter were not smooth because only predictions at time of observations were reported. We have corrected the figures accordingly (see Figure 2b and Figure S8 in Supplementary Materials).

Reviewer #2 (Recommendations for the authors):– What the authors state the main contribution as (new framework of modeling) and what I found to be the main conclusion (vaccine-induced antibody binding indicates protectiveness) are different. Because the model and analytical approach is not particularly novel, I would suggest redesigning the manuscript to highlight the data and biological conclusions.

We think that the use of such mechanistic modelling for defining a mechanistic correlate of protection constitutes a novelty because, to our knowledge, mechanistic CoP has been defined with counterfactual approaches only (6).

However, we do acknowledge that the important message here is the result in the context of COVID-19, therefore we have better highlighted the biological conclusion and dampen the methodological aspect (see section Introduction page 5, lines 77, 80-82 and section Discussion page 14, lines 289-290).

[Modification applied in section Introduction, page 5, line 77 and lines 80-82]

“Here, we propose to apply a model-based approach on NHP studies to evaluate i) […]. First, we present a mechanistic approach based on ordinary differential equation (ODE) models reflecting the virus-host interaction inspired from models proposed for SARS-CoV-2 infection (26–31) and other viruses (32-35).”

[Modification applied in section Discussion, page 14, lines 289-290]

“We explored the mechanistic effects of three SARS-CoV-2 vaccines and assessed the quality of markers as mechanistic CoP (mCoP).”

– Getting into the business of claiming there is some magical "correlate" is very tricky. I don't feel that this is what we need to understand the infection or response to vaccination, and the work in this manuscript does not assess sufficient data to make broad claims about any clinically-meaningful correlate. With as much data as there is in the literature, a validation using human data is warranted.

We certainly started this work with the hypothesis that neutralization is a likely candidate. However, in this work we try to respond to the following unknown questions:

– How much the neutralization could explain the dynamics of viral load?

– Which neutralization assay best explain the dynamics of the viral load?

– Is there any additional marker that participates to the control of the viral load and how it works?

The validation with human data is not straightforward as the quantification of the effects obtained in this study is helped by the experimental conditions of the studies in NHP: the date of infection is known and the viral load is precisely tracked.

As recommended, we have modified the manuscript to focus more on the results in these NHP experiments than the framework for defining a mCoP (see section Conclusion, page 17, lines 358-360)

[Addition applied in section Conclusion, page 17, lines 358-360]

“In conclusion, the modelling of the response to two new promising SARS-CoV-2 vaccines in NHP revealed a combination of effects with a blockade of new cell infections and the destruction of infected cells.”

– In the introduction, the authors reference studies that suggest "binding antibodies to SARS-CoV-2 and in vitro neutralization of virus infection are clearly associated with protection". This seems to be the same conclusion that the authors came to and makes me question: what is new here?

Two original aspects should be seen here:

1. We provide an indication on the biological effect of markers suspected as correlates of protection

2. We provide a quantification of the effect of the vaccine and more precisely of the markers that play the role of surrogate markers

– A more robust introduction that highlights the current conclusions and limitations of models in the literature is needed, particularly because the model presented is only a minor modification of those models. In addition, the model lacks immune dynamics, which would make its usefulness limited.

We have revised the Discussion section to better underline the existing literature on such dynamical models. We have also included a point on the limitation due to absence of modelling of the dynamics of the immune response (see Introduction, page 5, lines 80-82 and section Discussion, pages 16-17, lines 342-351 and 356-358).

[Modification applied in section Introduction, page 5, lines 80-82]

“First, we present a mechanistic approach based on ordinary differential equation (ODE) models reflecting the virus-host interaction inspired from models proposed for SARS-Cov-2 infection (26–31) and other viruses (32-35).”

[Modification applied in section Discussion, pages 16-17, lines 340-349 and 354-356]

“This result needs to be confirmed as it may also be the consequence of a different local immune response (60). The choice of the structural model defining the host-pathogen interaction is a fundamental step in the presented approach. Here, it was well guided by the biological knowledge, the existing models for viral dynamics (34, 61, 62) and the statistical inference allowing the selection of the model that best fit the data. Nevertheless, many modeling choices for the statistical model were made in this approach and more theoretical work evaluating the robustness of the results in their regards may be relevant for future works. In particular, we could relax the constraint of linear interpolation of marker dynamics by using simple regression models, allowing in the same time the integration of error model to account for measurement error for time-varying covariates (63–65). […]. Finally, it should be underlined that the dynamics of the immune response has not been modelled as suggested for instance for B cell response (68). This clearly constitutes the next step after the selection of the markers of interest as done in the present work.”

– In addition to the point above, while the model is a standard, previously published viral dynamics model with minor modifications, it was not adequately described for the broad readership of eLife. Perhaps I missed it, but I couldn't find where the terms were statistically justified?

Many corrections were made in the manuscript to better justify our modelling choices. The reviewer can refer to the comments (1a), (1b) and (3a) of the Reviewer 1.

We also added justifications about our modeling choices for the statistical model describing the three parameters β, δ and *P:*

(a) A log-transformation was considered for δ and *P* to ensure their positivity while a log10-transformation was used for β to also tackle some convergence issues and enable a good estimation of the fisher matrix.

(b) The two URT compartments were jointly considered in the statistical model for the three parameters. Although these latter can take different values between the compartments, this modeling choice allowed for the sharing of random effects and covariates. This choice was driven by the hypothesis of homogenous inter-individual variability within the URT. Moreover, this modeling of the parameters appeared us as more mechanistically correct rather than considering totally distinct parameters, and provided more power for their estimation.

To account for these different points, the manuscript was modified as follows (see section Methods page 23-24, Lines 487-506). Furthermore, to better explain our final choice for the statistical model we added an additional file, entitled “Model Building” describing the procedure applied.

Finally, we added a discussion in the manuscript about how these modeling choices could influence the selection of covariates (see section Discussion, page 16, Lines 349-353).

[Addition applied in section Methods, page 23-24, Lines 487-506]

“A log-transformation was adopted for the parameters δ and *P* to ensure their positivity while a log_10_-transformation was chosen for viral infectivity to also improve the convergence of the estimation. […] Finally, the adjustment of the model for the categorical covariates of groups of treatment, natural infection and/or vaccination, identified β and *δ* as the parameters with a statistically significant effect of these covariates (i.e., φ^P^_conv_ = 0 and φ^P^_CD40_=0).”

[Addition applied in section Discussion, page 16, Lines 349-353]

“…measurement error for time-varying covariates. Moreover, by construction, we assumed similar interindividual variability and effects of covariates within the two URT compartments as well as similar values for the viral infectivity and the loss rate of infected cells. Viral load dynamics measured in lungs being different from those in the URT (66, 67), the relaxation of this hypothesis of homogeneous physiological behavior in the URT may be pertinent to extend the model to the LRT. Finally, it should be underlined that the dynamics of the immune response ….”

– Several models have shown that viral load data, even when defined between compartments, is insufficient to distinguish transport between the nasal passages and trachea (e.g., Khan et al. Viruses, Ke et al. MedRxiv, Pinky et al. PCB, among others). In light of these studies, this part of the model seems unnecessary and one unsupported by the data. In addition, these other studies have also shown that viral and immune parameters are distinct in the nasopharynx and trachea.

We thank the reviewer for these relevant comments and the papers. Considering this remark and the remark 2 of the Reviewer 1, we modified the Results section to integrate papers mentioned in this comment and we transferred the section in which we tried to quantify virus exchanges in Appendix 1 “Model building”. Moreover, we added in this file the procedure applied on our model to build our statistical model and identify similar viral infectivity and loss rate of infected between the trachea and the nasopharynx. Nevertheless, although parameters were jointly estimated between the two compartments, distinct values of viral production were found.

In the authors' work, the data were joined and no effective comparison was made, which may cloud the parameter estimates and conclusions. Minimally, a discussion and reference of these published works is needed.

The parameter estimates obtained in the first study with the model adjusted for the groups of intervention were compared with previously reported modelling results. This comparison was included in the Results section (see pages 7-8, Lines 139-158).

[Modification / addition applied in section Results, pages 7-8, Lines 139-158]

“We estimated the viral infectivity at 0.95x10^-6^ (CI_95%_ [0.18x10^-6^; 4.94x10^-6^]) (copies/ml)^-1^ day^-1^ in naïve animals, which is in the range of previously reported modelling results whether in the case of SARS-CoV-2 virus (27, 29) or influenza (32, 33). […] In particular, they are in the range of estimates obtained within the URT, either in NHP (28) or in humans (29), with the product pxT_0_ equals to 15.1x10^8^ (CI_95%_ [3.98x10^8^; 58.1x10^8^]) and 0.21x10^8^ (CI_95%_ [0.088x10^8^; 0.48x10^8^]) virions/ml/day in the nasopharynx and the trachea, respectively.…”

– Another limitation of the model and the data is that viral loads in the nasopharynx can be similar when disease is not. Can the approach here be used to assess vaccine efficacy (in terms of reducing disease)?

The mathematical model can be extended to incorporate clinical progression as it has been proposed in other contexts (8). However, it was not possible to fit such model here because of the reduce sample size of the first trial and clinical data were not available in the two others.

– Curiously, CD8 T cells were measured in the serum, but this type of measurement tends to not reflect the events in the tissue where resident T cells are thought to be important. There were some comments made about the infected cell clearance rate, but it did not seem that the data were used to evaluate the parameter estimates or model conclusions.

In agreement to the reviewer comment, a great CD8 T cell response to the αCD40.RBD vaccine has been demonstrated in mice splenocytes but was not found in macaques’ blood (9). We systematically analyzed the available markers but we were not surprised that we could not estimate an association between the CD8 T cell response in the blood and the clearance of infected cells. This is what we have quickly discussed in the current paper.

– Macaques, unfortunately, are not a great model to assess the individual heterogeneity observed in humans. Dissecting the heterogeneity from different sources would be greatly beneficial, but it seems naïve to assume that this can be done using animal models. I suggest using a human dataset, where there are many to choose from in the literature, or artificially creating one from human data and see if the model can still perform.

It is true that results may differ from macaques and humans although the importance of neutralizing antibodies has been demonstrated in the two species. One advantage of the experimentations in macaques was the availability of repeated measures of viral load and the control of the date of infection which were very helpful for the estimation of model parameters. The translation of such model in humans is certainly an interesting perspective that we are now mentioning in the Discussion section (see page 17, lines 368-370).

[Addition applied in section Discussion, page 17, Lines 368-370]

“… waning protective antibody levels (71, 72), at least in NHP studies although the framework could be extended to human studies using mixed approaches of within and between hosts modelling (73) providing that enough information is collected.”

– In reference to Table S1, it is stated that vaccinated and unvaccinated animals could be distinguished. Why is this the goal? We would know who is/isn't vaccinated – a better question is who among the vaccinated would not be protected from infection and/or disease.

The objective was not to distinguish which animal was vaccinated or not but to use this information to identify which part of the model would be impacted and thus on which biological mechanisms vaccination and/or natural infection seem to play a role. In the Table S1, information about groups is used for the criterion 2 to verify that the immune marker selected by the selection method is able to capture information provided by the group: once adjusted for the maker, the covariate of group is no more significant.

– In Line 189-191, the authors state "both specific antibodies and specific CD8^+^ T cells are mechanisms commonly considered important for killing infected cells. We retained the anti-RBD binding IgG Ab that were positively associated to the increase of the loss of infected cells.". Unfortunately, this statement is incorrect as antibodies cannot kill infected cells. Antibodies neutralize virus so that cells are not infected, but they cannot kill cells.

The antibodies can prevent target cell infection and concur to the opsonization of viral particles but there are also playing an important role for the destruction of infected cells by several (Fc-mediated) pathways including antibody-dependent cellular phagocytosis, antibody-dependent cellular cytotoxicity or complement-dependent cytotoxicity (see for instance Figure 2 in (11)).

– Prior studies have found that antibodies have a limited role during infection (e.g., see Goyal et al. Viruses). How does this fit into the results?

It is true that cellular response is expected to play a greater role during infection. However, antibodies are still playing an important role to control ongoing infection as demonstrated by neutralizing monoclonal antibodies used for treatment of COVID-19 (see (11)).

– Figure 2 is difficult to assess what is data and what is the model, and the model fit is difficult to see. It seems a though the 95% confidence intervals (gray shading) are also not indicative the model. Why are there multiple peaks and does the model capture that?

We thank the reviewer for this relevant comment. The reviewer can refer to the comment 7 of the reviewer 1 for our answer about this figure. A new version of Figure 2 is now proposed.

– The discussion could be improved. It lacked discussion of how the work fits into the literature, particularly other published SARS-CoV-2 models, how the correlate might be used in the clinic, etc. In addition, a discussion about the differences in dynamics within the nasopharynx and trachea in vaccinated individuals would have been interesting.

The Discussion section has been revised by adding:

– the perspective on extension of the model inspired by other models proposed in the literature

– one more sentence on the differences in dynamics within the nasopharynx and trachea

– a perspective on clinical implications

[Addition/modification applied in section Discussion]

The reviewer can refer to the comments 4, 5 and 10 of the reviewer 2 for the related modifications of the section Discussion.

– It's unclear to me why the reproduction number R was under 1 for the alphaCD40.RBD group when there was replication for several days in this group.

The reproductive ratio is reflecting the trend of the replication of the virus which is tending to extinct over time when R<1. However, at a given time, R could be < 1 although there are still infected cell and replicating virus. R is estimated to be <1 in the nasopharynx from the beginning and drop below 1 in the trachea from day 2. Hence, the dynamics of the observed gRNA is a contribution of the virus inoculated in the first days and some replication (below the detection threshold in the nasopharynx and with one observed blip in the trachea).

Reviewer #3 (Recommendations for the authors):First, a suggestion regarding my overfitting concern:• The number of non-human primates in each arm of these studies is understandably limited, but cross-validation could be used to reduce the extent of overfitting even with limited replicates.

We thank the reviewer for this relevant comment. Indeed, the number of NHPs is quite limited. As recommended, we performed a bootstrap procedure with replacement (B=50 times) to obtain more robust estimates. Due to time constraints, the procedure was performed only for the model adjusted for group of treatment and the one with β adjusted the selected marker. The bootstrap parameter estimate was calculated as the median of the parameter estimates from the B bootstrap samples while the standard error of each parameter was calculated according to the definition of Thai et al. (2013) (12), which means with the SE of the *l*-th component of the vector of parameters given by:SE^B(l)=1B−1∑b=1B(θ^b∗(l)−θ^B(l))2 with θ^b∗(l) being its estimate obtained at the *b*-th iteration of the bootstrap and θ^B(l) the bootstrap parameter estimate. For each bootstrap sample, we paid attention to keep the 1:1:1 ratio between the 3 groups of treatment, with 6 animals selected within each group. Results were then reported in Table 1 and Table 2 below**,** respectively for the two models aforementioned.

In addition, we performed a convergence assessment in Monolix for these two models to evaluate the robustness of the convergence, with 15 runs. The results are shown in Author response image 1 and Author response image 2.

**Author response image 1. sa2fig1:** Results of the convergence assessment performed on the model with both β and δ adjusted for the group of treatment. Fifteen iterations were performed to evaluate the robustness of the estimation according to the initial value of the parameters.

**Author response image 2. sa2fig2:** Results of the convergence assessment performed on the model with β adjusted for ACE2-RBD binding inhibition (ECL RBD). Fifteen iterations were performed to evaluate the robustness of the estimation according to the initial value of the parameters.

[Addition applied in section Methods page 27, Lines 578-580]

“Furthermore, we performed a bootstrap procedure with replacement (79) (50 samples) on the optimal model to obtain and verify the robustness of the estimates (see Supplementary file 2 for the results), and to reduce potential overfitting that could result from the small number of NHPs.”

[Addition/modification applied in supplementary information file – Table S2 (now defined as Supplementary file 2)]

[Addition in the supplementary files of the Table S5 (Supplementary file 4) presenting parameter estimates when viral infectivity is adjusted for time-varying pseudo-neutralization (ECLRDB) and loss rate of infected cells adjusted for group effects]

Next, two suggestions to enhance clarity:• The rationale for the use of profile likelihood should be made explicit. Why do a few parameters using this approach and then later do rest with full maximum likelihood? Is it a limitation of the optimization software?

As mentioned by the Reviewer 1 in the comment (3b), we performed profile likelihood to determine the value of the three fixed parameters *c*, *c_i_* and *k*. We used profile likelihood for these parameters because of identifiability issues. Similar to Gonçalves et al. (2021) (3), data did not provide enough information to estimate the duration of the eclipse phase and virus clearances with full maximum likelihood (e.g., not enough time points or data not enough specific to the mechanisms of interest). Profile likelihood (also referred as sensitive analyses in some papers) was then performed to roughly identify values that could be biologically relevant and lead to better estimations of the model. Other parameters are then estimated considering *c*, *c_i_* and *k* fixed at the values found by profile likelihood. The reviewer can refer to comments (3a) and (3b) to see the modifications we applied to the manuscript to better explain this point.

• In the model, why separately track virions from the inoculum versus virions generated in host? If it is just to allow for different clearance rates, why do we expect these to be different?

We distinguished virions that have been inoculated and those produced by infected cells. The reviewer can refer to the comment 1. (b) of reviewer 1 for our feedback about this comment.

References

1. Czuppon P, Débarre F, Gonçalves A, Tenaillon O, Perelson AS, Guedj J, et al. Success of prophylactic antiviral therapy for SARS-CoV-2: Predicted critical efficacies and impact of different drug-specific mechanisms of action. PLOS Computational Biology. 1 mars 2021;17(3):e1008752.

2. Krambrich J, Akaberi D, Ling J, Hoffman T, Svensson L, Hagbom M, et al. SARS-CoV-2 in hospital indoor environments is predominantly non-infectious. Virology Journal. 2 juin 2021;18(1):109.

3. Gonçalves A, Maisonnasse P, Donati F, Albert M, Behillil S, Contreras V, et al. SARS-CoV-2 viral dynamics in non-human primates. PLOS Computational Biology. 17 mars 2021;17(3):e1008785.

4. Sawicki SG, Sawicki DL, Siddell SG. A Contemporary View of Coronavirus Transcription. Journal of Virology. janv 2007;81(1):20‑9.

5. Miao H, Xia X, Perelson AS, Wu H. On Identifiability of Nonlinear ODE Models and Applications in Viral Dynamics. SIAM Review. janv 2011;53(1):3‑39.

6. Gilbert PB, Fong Y, Carone M. Assessment of Immune Correlates of Protection via Controlled Vaccine Efficacy and Controlled Risk. arXiv:210705734 [stat] [Internet]. 12 juill 2021 [cité 8 févr 2022]; Disponible sur: http://arxiv.org/abs/2107.05734

7. Robinot R, Hubert M, de Melo GD, Lazarini F, Bruel T, Smith N, et al. SARS-CoV-2 infection induces the dedifferentiation of multiciliated cells and impairs mucociliary clearance. Nature communications. 2021;12(1):4354.

8. Guedj J, Thiébaut R, Commenges D. Joint Modeling of the Clinical Progression and of the Biomarkers’ Dynamics Using a Mechanistic Model. Biometrics. 2011;67(1):59‑66.

9. Marlin R, Godot V, Cardinaud S, Galhaut M, Coleon S, Zurawski S, et al. Targeting SARS-CoV-2 receptor-binding domain to cells expressing CD40 improves protection to infection in convalescent macaques. Nat Commun. 1 sept 2021;12(1):5215.

10. Goyal A, Reeves DB, Schiffer JT. Multi-scale modelling reveals that early super-spreader events are a likely contributor to novel variant predominance. Journal of The Royal Society Interface. 19(189):20210811.

11. Taylor PC, Adams AC, Hufford MM, de la Torre I, Winthrop K, Gottlieb RL. Neutralizing monoclonal antibodies for treatment of COVID-19. Nat Rev Immunol. juin 2021;21(6):382‑93.

12. Thai HT, Mentré F, Holford NHG, Veyrat-Follet C, Comets E. Evaluation of bootstrap methods for estimating uncertainty of parameters in nonlinear mixed-effects models: a simulation study in population pharmacokinetics. Journal of Pharmacokinetics and Pharmacodynamics. févr 2014;41(1):15‑33.